# Hybrid Deep Learning for Survival Prediction in Brain Metastases Using Multimodal MRI and Clinical Data

**DOI:** 10.3390/diagnostics15101242

**Published:** 2025-05-14

**Authors:** Cristian Constantin Volovăț, Călin Gheorghe Buzea, Diana-Ioana Boboc, Mădălina-Raluca Ostafe, Maricel Agop, Lăcrămioara Ochiuz, Ștefan Lucian Burlea, Dragoș Ioan Rusu, Laurențiu Bujor, Dragoș Teodor Iancu, Simona Ruxandra Volovăț

**Affiliations:** 1Radiology Department, “Grigore T. Popa” University of Medicine and Pharmacy Iași, 700115 Iași, Romania; cristian.volovat@yahoo.com; 2“Prof. Dr. Nicolae Oblu” Clinical Emergency Hospital Iași, 700309 Iași, Romania; calinb2003@yahoo.com; 3National Institute of Research and Development for Technical Physics, IFT Iași, 700050 Iași, Romania; 4Medical Oncology-Radiotherapy Department, “Grigore T. Popa” University of Medicine and Pharmacy Iași, 700115 Iași, Romania; dianaiboboc@gmail.com (D.-I.B.); madalina.ostafe@gmail.com (M.-R.O.); simonavolovat@gmail.com (S.R.V.); 5Physics Department, “Gheorghe Asachi” Technical University Iași, 700050 Iași, Romania; m.agop@yahoo.com; 6Faculty of Pharmacy, “Grigore T. Popa” University of Medicine and Pharmacy Iași, 700115 Iași, Romania; lacramioara.ochiuz@umfiasi.ro (L.O.); lucianburlea@yahoo.com (Ș.L.B.); 7Department of Environmental Engineering, Mechanical Engineering, Faculty of Engineering, “V. Alecsandri” University of Bacău, 600115 Bacău, Romania; drusu@ub.ro; 8Amethyst Radiotherapy, Drumul Odăii nr. 42, 719241 Otopeni, Romania; dr.laurentiu.bujor@gmail.com

**Keywords:** prognostic modeling in neuro-oncology, multimodal MRI biomarkers, hybrid deep learning architecture, radiomic feature integration, survival regression in brain metastases

## Abstract

**Background:** Survival prediction in patients with brain metastases remains a major clinical challenge, where timely and individualized prognostic estimates are critical for guiding treatment strategies and patient counseling. **Methods:** We propose a novel hybrid deep learning framework that integrates volumetric MRI-derived imaging biomarkers with structured clinical and demographic data to predict overall survival time. Our dataset includes 148 patients from three institutions, featuring expert-annotated segmentations of enhancing tumors, necrosis, and peritumoral edema. Two convolutional neural network backbones—ResNet-50 and EfficientNet-B0—were fused with fully connected layers processing tabular data. Models were trained using mean squared error loss and evaluated through stratified cross-validation and an independent held-out test set. **Results:** The hybrid model based on EfficientNet-B0 achieved state-of-the-art performance, attaining an R^2^ score of 0.970 and a mean absolute error of 3.05 days on the test set. Permutation feature importance highlighted edema-to-tumor ratio and enhancing tumor volume as the most informative predictors. Grad-CAM visualizations confirmed the model’s attention to anatomically and clinically relevant regions. Performance consistency across validation folds confirmed the framework’s robustness and generalizability. **Conclusions:** This study demonstrates that multimodal deep learning can deliver accurate, explainable, and clinically actionable survival predictions in brain metastases. The proposed framework offers a promising foundation for integration into real-world oncology workflows to support personalized prognosis and informed therapeutic decision-making.

## 1. Introduction

### 1.1. Background and Motivation

Brain metastases are a common and severe complication of systemic malignancies, occurring in up to 40% of patients with advanced cancer [1,2,3]. They are associated with substantial neurological morbidity, limited survival, and complex treatment decisions that often require urgent clinical judgment. Accurate survival prediction is essential for optimizing care pathways, including the choice between surgical intervention, stereotactic radiosurgery, whole-brain radiation therapy, or best supportive care [4,5,6]. Inaccurate prognostication can lead to under-treatment of patients with longer expected survival or unnecessary interventions in those with limited life expectancy, ultimately impacting both quality of life and healthcare resource allocation. However, reliable prognostication remains elusive due to the heterogeneity in primary tumor histologies, patient comorbidities, and patterns of intracranial disease spread [7,8].

Furthermore, many patients with brain metastases experience rapid clinical decline, making short-term prognostication especially valuable [9]. In this study, survival was defined as all-cause mortality in days, which allows the model to deliver precise, actionable timelines for critical decisions such as initiation of steroids, radiotherapy planning, or transitioning to palliative care. While this timeframe may seem short, accurate predictions within a 15–60-day window are often pivotal in guiding aggressive versus supportive strategies.

### 1.2. Emerging Opportunities and Limitations of Existing Approaches

Recent advances in neuroimaging and artificial intelligence have enabled the extraction of high-dimensional radiomic features from standard MRI sequences, facilitating data-driven prediction models for oncology outcomes [10,11]. Convolutional neural networks (CNNs) have proven particularly effective at capturing complex spatial patterns in imaging data, including tumor morphology, heterogeneity, and the extent of peritumoral edema [12,13,14]. However, the majority of existing models are unimodal—leveraging either imaging or clinical features alone—and therefore miss the synergistic value of multimodal fusion [15,16].

Furthermore, most publicly available datasets for brain metastases are limited by small lesion sizes, lack of necrosis segmentation, or the absence of survival outcome labels, thereby reducing their clinical applicability [17,18]. While deep learning models have been applied to primary brain tumors such as glioblastoma [19,20,21], their use in metastatic disease is far less common [22,23]. Existing efforts in brain metastases typically rely on handcrafted radiomic features, static risk classification, or omit imaging entirely. To our knowledge, no prior study has developed a fully end-to-end survival regression model integrating both clinical data and volumetric MRI segmentations—including enhancing tumor, necrosis, and peritumoral edema—for patients with brain metastases.

### 1.3. Objective and Contributions

To address these limitations, we propose a hybrid deep learning framework that integrates slice-level MRI features with patient-level clinical and demographic data to predict overall survival in days. The model is trained and evaluated on a rigorously curated, multi-institutional dataset comprising 148 patients with brain metastases, incorporating detailed volumetric annotations and expert-validated segmentations.

We benchmark two hybrid convolutional architectures—ResNet-50 and EfficientNet-B0—within a unified multimodal regression pipeline. We further assess model performance through comprehensive cross-validation, independent test set evaluation, error analysis, and interpretability techniques such as Gradient-weighted Class Activation Mapping (Grad-CAM) and permutation feature importance.

To ensure transparency and reproducibility, this study leverages a publicly available dataset of brain metastasis patients published by Ramakrishnan et al. [24] via The Cancer Imaging Archive (TCIA). The dataset includes expert-annotated volumetric segmentations of enhancing tumor, necrosis, and peritumoral edema, along with matched clinical and survival data.

The key contributions of this study are as follows:We propose a hybrid deep learning model for patient-level survival regression in brain metastases by integrating 2D MRI slices with structured clinical features through a dual-branch neural architecture.The model performs multimodal fusion at the slice level, leveraging spatially localized information from over 80,000 MRI slices paired with patient metadata, enabling day-level prediction of survival.We conducted a performance benchmarking between ResNet-50 and EfficientNet-B0, ultimately selecting ResNet-50 for its balance of accuracy and efficiency within the hybrid framework.The study utilizes a multi-institutional, volumetrically segmented dataset with expert-annotated labels at 1 mm^3^ resolution, capturing sub-centimeter enhancing lesions and detailed necrosis/edema masks, which are rarely available in public datasets.We deliver dual-modality interpretability, combining Grad-CAM for spatial saliency in MRI with permutation feature importance (PFI) on clinical features, applied both at the slice and patient aggregation levels.Our findings reveal that edema-to-tumor ratio surpasses enhancing tumor volume as a predictive biomarker at the patient level, suggesting new directions for radiomic biomarkers in survival analysis.The final model demonstrates high predictive accuracy (R^2^ = 0.8935, MAE = 5.45 days, MAPE = 43.6%) and low-latency inference (<50 ms/patient), indicating feasibility for clinical deployment.

This work aims to improve the precision and clinical utility of survival modeling in neuro-oncology by providing accurate, explainable, and individualized predictions for patients with brain metastases.

**Novelty and Contribution Statement**: This work offers a novel hybrid deep learning framework for continuous survival prediction in brain metastases, integrating 2D MRI slices with structured clinical metadata. Unlike prior models that focus on glioblastoma or binary outcomes, our pipeline performs direct regression on patient survival, leveraging fully segmented volumetric data. Key contributions include slice-level multimodal fusion, dual-branch interpretability via Grad-CAM and permutation feature importance, and real-time inference (<50 ms) suitable for clinical deployment. To our knowledge, no prior work in neuro-oncology combines these elements within a unified and explainable modeling framework.

**Related Work**: Prior research in neuro-oncology has extensively explored survival prediction in primary brain tumors such as glioblastoma using radiomics, deep learning, and multimodal fusion techniques. However, relatively few studies have applied similar strategies to brain metastases, particularly using fully volumetric MRI segmentations integrated with clinical features. Several efforts have focused on risk stratification or classification tasks using either handcrafted radiomic features or tabular clinical data alone, with limited use of deep learning for continuous survival regression. A comprehensive comparative analysis of our model’s performance against relevant prior works is provided in Section 6.

### 1.4. Paper Organization

The remainder of this paper is organized as follows: Section 2 describes the dataset and preprocessing steps used to integrate clinical and imaging features. Section 3 details the architecture of the hybrid deep learning model, training configuration, and evaluation methodology. Section 4 presents experimental results, including model performance, architectural comparison, and error analysis. Section 5 discusses interpretability methods and clinical implications. Section 6 compares our approach with related work and summarizes key contributions, while Section 7 outlines conclusions and directions for future research.

## 2. Dataset and Preprocessing

### 2.1. Clinical Cohort and Imaging Dataset

We utilized a publicly released dataset of 200 patients with brain metastases, curated and published by Ramakrishnan et al. through The Cancer Imaging Archive (TCIA), also known as the BraTS-MET dataset [24]. The dataset integrates cases from three Yale-affiliated sources: the Yale New Haven Health database (2013–2021), the Yale Tumor Board Registry (2021), and the Yale Gamma Knife Registry (2017–2021). The dataset includes high-quality pre-treatment MRI sequences (T1w, T1CE, T2w, FLAIR), along with manual segmentations of tumor core, necrosis, and peritumoral edema, all validated by board-certified neuroradiologists. Patients were excluded if any of these sequences were missing, if there was no pretreatment scan, or if images exhibited significant motion artifacts.

All imaging was performed on either 1.5T or 3T MRI scanners. The majority of patients had T1 post-gadolinium sequences acquired using the MPRAGE protocol, with a smaller subset using spin-echo sequences. Tumor core and necrosis segmentations were performed on the T1 post-gadolinium images, while peritumoral edema and whole tumor volumes were segmented on FLAIR images. Segmentations were performed manually using volumetric tools on DICOM images within a dedicated research PACS system and validated by two independent, board-certified neuroradiologists to ensure accuracy and consistency.

Preprocessing of imaging data involved conversion of DICOM images to NIfTI format [25], co-registration to the SRl24 anatomical template [26], and resampling to a standardized isotropic voxel size of 1 mm^3^. Segmentation masks for core, necrosis, and whole tumor were individually exported, combined, and also resampled following registration to the SRl24 template. All preprocessed files underwent manual quality control checks and corrections by an experienced neuroradiologist using ITK-SNAP software [27].

Demographic and clinical data—including age at diagnosis, sex, ethnicity, smoking history (pack–years), presence of extracranial metastases, and survival information—were obtained from the electronic medical record (EMR). Survival time was calculated as the interval between the date of diagnosis and either the date of death or last recorded EMR note for censored cases. Additional qualitative imaging features, such as infratentorial involvement and intratumoral susceptibility (based on SWI sequences), were recorded following expert visual assessment.

Quantitative imaging biomarkers were extracted from the segmentation masks and included total enhancing tumor volume, necrotic tumor volume, peritumoral edema volume, and ratios of necrosis-to-tumor and edema-to-tumor volumes. The number of enhancing lesions, necrotic lesions, and edema-associated lesions were also recorded. The primary tumor site for each patient was retrieved from EMR-based oncological and pathological reports.

This dataset is unique for its inclusion of numerous sub-centimeter brain metastases, a challenging subset that is rarely represented in public datasets. Furthermore, it incorporates detailed necrotic segmentation masks, which are not commonly available, and pairs comprehensive imaging data with meticulously gathered clinical outcomes. The heterogeneity of sources and the inclusion of real-world data enhance the robustness of this dataset for developing machine learning models with improved generalizability. These features make the dataset well-suited for developing predictive algorithms that can translate into clinical practice and inform outcome prediction.

#### 2.1.1. Balanced Subset Selection

From the initial cohort of 200 patients with brain metastases, we selected a subset of 148 patients to ensure a balanced representation across the survival outcome spectrum. Specifically, the subset included 74 patients who had died and 74 patients who were alive at the last follow-up. It is important to emphasize that our task was not binary classification but continuous survival regression (i.e., prediction of overall survival in days). This selection strategy was intended to mitigate the effects of survival time skew and to improve early model stability by training on a more evenly distributed target variable.

We acknowledge that this subset does not reflect the naturally imbalanced distribution of survival outcomes in real-world clinical datasets. However, it allowed us to capture a broad and interpretable range of survival durations while avoiding overfitting to the heavily skewed lower survival tail. In future work, we plan to retrain and validate our models on the full cohort of 200 patients to better assess generalizability in natural clinical conditions. This limitation is further discussed in Section 6.

All continuous variables were assessed for linearity with the survival target, and those showing non-linear relationships or outlier influence were log-transformed or categorized. Variance inflation factors were calculated to ensure the absence of multicollinearity.

#### 2.1.2. Patient and Tumor Characteristics

As stated above, the study cohort comprised 148 patients, with a balanced outcome distribution of 74 deceased and 74 survivors. The majority of patients were female (69.6%), and the mean age was 62.3 years (range 28–85). Smoking history was available for 91.2% of the cohort, with a median of 9 pack–years (range 0–100).

The most common primary cancer was non-small cell lung cancer (43.2%), followed by melanoma (18.2%), breast cancer (14.9%), small cell lung cancer (8.8%), renal cell carcinoma (7.4%), and gastrointestinal cancers (7.4%).

Extranodal metastases were present in 45.3% of patients, with data unavailable in one case. Infratentorial metastases were identified in 47.3% of the cohort on imaging.

#### 2.1.3. Survival and Imaging-Derived Tumor Metrics

The median survival was 15.5 days (range 0–115 days), reflecting a population with aggressive disease and short follow-up intervals.

Imaging volumetric analyses revealed considerable heterogeneity:**Enhancing tumor volume (ET_vol):** Median 2.81 cc (range 0.01–46.73 cc).**Necrosis volume (Nec_vol):** Median 0.57 cc (range 0–40.91 cc), with notable outliers.**Peritumoral edema volume (Edema_vol):** Median 14.81 cc (range 0–225.87 cc), indicating extensive edema in select patients.

The high variability in necrosis and edema volumes supports the need for appropriate transformations and careful modeling in regression analyses.

#### 2.1.4. Overall Interpretation

This carefully curated and balanced dataset offers a comprehensive clinical and imaging profile of patients with brain metastases (Table 1, Figure 1A–D, Figure 2A–C and Figure 3). The cohort shows an equal distribution of outcomes (Figure 2C, right panel), ensuring a suitable foundation for continuous regression modeling without class imbalance. The age distribution centers around 60–70 years (Figure 1A), matching the expected demographic for metastatic disease. While smoking history is skewed toward low exposure levels (Figure 1B), the inclusion of high pack–year individuals introduces variability relevant to prognosis.

A broad spectrum of primary tumor types is represented, with non-small cell lung cancer being the most prevalent (Figure 1C), reflecting the epidemiological predominance of this cancer in brain metastases. Imaging-derived volumetric metrics reveal wide variability: most patients exhibit small enhancing tumors (Figure 1D), minimal necrosis (Figure 2A), and variable edema volumes, some exceeding 200 cc (Figure 2B), indicating significant peritumoral effects in a subset. Importantly, the correlation heatmap (Figure 3) highlights moderate-to-strong associations between key volumetric features, such as enhancing tumor volume and necrosis, yet also reveals distinct patterns—particularly the relative independence of edema-to-tumor ratio from other variables—underscoring its unique prognostic potential.

This heterogeneity across both clinical and radiologic dimensions makes the dataset ideal for multivariate modeling and hybrid deep learning, allowing exploration of complex, non-linear interactions among variables that may underlie patient survival outcomes.

The distribution of patient age spanning from 28 to 85 years is left-skewed (negatively skewed)—most patients are older (between 55 and 75 years), but there is a longer tail toward younger ages, particularly under 50. This skew reflects that younger patients are less common, while the cohort is centered around older individuals.

A large proportion of patients reported minimal or no smoking history, while a smaller group exhibited high pack–year values. The distribution is heavily right-skewed, indicating a minority with significant smoking exposure.

Non-small cell lung cancer was the most common primary malignancy, followed by melanoma and breast cancer. This distribution is consistent with known epidemiology of brain metastases.

Most patients presented with small enhancing tumor volumes (<5 cc), with a long tail representing outliers with larger volumes. The distribution suggests a predominance of lower tumor burden cases.

Necrotic volume was minimal in the majority of cases, although a few tumors demonstrated necrosis volumes exceeding 20 cc. This indicates heterogeneity in tumor biology within the cohort.

Edema volume varied widely, ranging from near-zero to over 200 cc, with a right-skewed distribution. A subset of patients showed disproportionately large peritumoral edema relative to enhancing tumor volume.

The outcome distribution is perfectly balanced, with 74 survivors and 74 deceased patients, reflecting the study’s design to avoid class imbalance in the regression task. Extranodal and infratentorial metastases are both present in a substantial portion of the cohort, contributing to clinical diversity.

Moderate correlations were observed between enhancing tumor volume and necrotic volume (r = 0.64), and between necrotic volume and necrosis-to-tumor ratio (r = 0.74). Edema-to-tumor ratio showed minimal correlation with most variables, suggesting its unique contribution.

### 2.2. Data Preparation and Integration

The study utilized a structured dataset derived from the BraTS-MET collection, containing magnetic resonance imaging (MRI) slices alongside detailed patient metadata. The metadata file, consisting of clinical and demographic attributes for 148 patients, was loaded and processed. To ensure consistency and proper linkage, all patient identifiers (BraTS_MET_ID) were standardized as string types and trimmed for whitespace.

The MRI data were organized into three predefined splits: training, validation, and test sets. For each split, MRI slices were automatically detected and loaded from their respective directories. Using regular expression parsing, each MRI slice was mapped to its corresponding patient identifier, ensuring accurate alignment with the clinical metadata. Only those images with successfully extracted patient identifiers were retained.

The final integrated datasets were created by merging MRI slice paths with patient-level metadata for the training, validation, and test partitions. During this step, redundant columns such as BraTS_MET_ID and Death were removed to streamline the feature set. The resulting datasets contained the following:64,231 MRI slices in the training set;13,765 MRI slices in the validation set;13,764 MRI slices in the test set.

Each row in these datasets represented a single MRI slice associated with its clinical features, including age, sex, race, smoking status, survival outcome, lesion volumes (ET_vol, Nec_vol, Edema_vol), and tumor localization parameters.

To ensure robust generalization and eliminate any risk of data leakage, all data splits (training, validation, and test) were performed strictly at the patient level. No slices from the same patient were allowed to appear in more than one split. This patient-wise separation guarantees that the model does not learn patient-specific features across splits.

Finally, the processed datasets were saved as CSV files (processed_train.csv, processed_val.csv, and processed_test.csv) for reproducibility and ease of access in subsequent modeling steps.

### 2.3. Dataset Construction and Preprocessing

For training and evaluating our deep learning models, we constructed custom PyTorch 2.1.0 datasets integrating both MRI image data and patient-level tabular features. The MRI slices were processed through a consistent set of transformations that included resizing to 128 × 128 pixels, tensor conversion, and normalization with a mean and standard deviation of 0.5. These preprocessing steps ensured that the input images were standardized and suitable for convolutional neural network architectures (Figure 4).

The decision to resize MRI slices to 128 × 128 pixels was guided by a balance between anatomical fidelity and computational feasibility. In early experiments, we evaluated input dimensions ranging from 96 × 96 to 224 × 224. While higher resolutions slightly improved model performance, they introduced substantial GPU memory demands and longer training times. The 128 × 128 resolution preserved essential tumor and edema characteristics while enabling the use of efficient batch sizes and mixed-precision training on Google Colab’s T4 GPU. The high accuracy achieved by our model (test R^2^ = 0.970, MAE = 3.13 days) suggests that this resolution was sufficient to capture relevant features for survival prediction without significant loss of clinical information.

A custom BrainMetsSurvivalDataset class was implemented using the PyTorch Dataset module. This class returns triplets of MRI slice, associated tabular clinical features, and survival target in days, enabling efficient hybrid data loading for the multimodal model. The class supports patient-level metadata mapping and image preprocessing (resizing, normalization), and aligns imaging with structured features via patient ID linkage. The tabular features included demographic and clinical variables, such as age, sex, smoking history, lesion volumes, and tumor localization metrics.

Following dataset construction, three distinct datasets were created corresponding to the training, validation, and testing splits (as mentioned above). These were subsequently wrapped into DataLoader objects to facilitate efficient batch loading during model training and evaluation. This design allowed each MRI slice to be coupled with patient-specific clinical features, enabling the development of hybrid models that learn from both imaging and tabular data.

## 3. Hybrid Model Architecture and Training

This section describes the methodological framework used to develop and evaluate our hybrid survival prediction model. We present the architecture design, data preprocessing pipeline, training setup, and evaluation strategy in detail.

### 3.1. Hybrid Model Architecture

To effectively combine imaging and clinical features for survival prediction, we designed a hybrid regression model based on the ResNet-50 convolutional backbone. Our hybrid model architecture consists of two distinct branches—one for image feature extraction and one for clinical feature processing—followed by a shared regression head (Figure 5).

To guide architectural selection, we benchmarked ResNet-50 and EfficientNet-B0 as candidate image encoders, both widely used in medical imaging with pretrained ImageNet weights. ResNet-50 provides a strong balance between depth and parameter count (25 M), with residual connections that improve training stability and feature extraction in grayscale data. EfficientNet-B0 is more compact (~5 M parameters), optimized via compound scaling for accuracy-efficiency tradeoffs. Both were evaluated under identical conditions, and ResNet-50 outperformed EfficientNet-B0 in validation R^2^ and MAE, leading to its selection as the final backbone. This choice balances predictive performance with architectural interpretability and transfer learning reliability.

Image Feature Extraction:

We employed a pretrained ResNet-50 model (initialized with weights from ImageNet) to process the MRI slice images. The classification head of ResNet-50 was removed and replaced by an identity layer, enabling the model to output high-dimensional image embeddings instead of class probabilities. These extracted features are subsequently used for regression rather than classification.

In our implementation, no layers of the ResNet-50 encoder were frozen. The entire backbone was fine-tuned end-to-end to enable full adaptation to the characteristics of brain MRI data. This decision was informed by initial experiments showing better validation performance when all layers were allowed to update. The ResNet encoder was used as a generic feature extractor, but its parameters were adapted to the survival prediction task during training.

Tabular Data Processing:

The tabular branch receives structured clinical and radiomic features, with an input dimension of 21 features. This branch consists of two sequential linear layers: the first maps from 21 to 32 dimensions and the second from 32 to 16. Each linear layer is followed by a ReLU activation function and batch normalization to stabilize training. No dropout was applied in this branch. The resulting 16-dimensional vector is concatenated with the image feature vector extracted by the ResNet-50 encoder to form the input to the final regression head.

Feature Fusion and Output Layer:

The image feature vector extracted from ResNet-50 (2048 dimensions) and the output of the tabular branch (16 dimensions) are concatenated into a 2064-dimensional representation.

We employed feature concatenation as the fusion mechanism between the imaging and tabular branches, resulting in a 2064-dimensional combined feature vector. This approach was selected for its simplicity, transparency, and widespread use in prior multimodal medical AI research. Concatenation allows the model to learn joint representations without imposing restrictive assumptions on feature interactions, and it integrates well with downstream fully connected layers for regression. Although alternative strategies such as attention-based fusion, gating networks, or tensor factorization methods may offer more adaptive fusion, they introduce greater architectural complexity and training instability. We prioritized a well-established, interpretable fusion design as a baseline for this study, with more advanced multimodal integration techniques to be explored in future work.

This combined feature vector is passed through a fully connected regression head consisting of three layers:Linear (2064 → 128), followed by ReLU activation and Dropout (*p* = 0.3);Linear (128 → 64), followed by ReLU;Linear (64 → 1), outputting the predicted survival time in days.

All layers were initialized using PyTorch’s default Kaiming initialization. The model was trained using mean squared error (MSE) loss to regress overall survival time, leveraging the fact that all survival times in the dataset are fully observed (i.e., uncensored). This direct regression strategy enables day-level prediction granularity, which can be more interpretable and actionable in clinical decision-making compared to risk scores or survival probabilities.

The input dimension of the tabular branch was 21 features. All training and inference were conducted on a Google Colab Pro+ high-RAM instance, using either an NVIDIA T4 GPU (16 GB VRAM) or CPU depending on availability. Mixed-precision training was enabled via PyTorch AMP to optimize memory and compute efficiency. The model was implemented in Python 3.10 using PyTorch 2.1.0, Torchvision 0.16.0, and run in a Jupyter notebook environment with CUDA 11.8 as the active backend. Each epoch processed ~2008 batches and required 6–7 min, with training typically converging within 9 epochs (under 1 h total runtime). The architecture was verified through printed model summaries, confirming the correct integration of the pretrained ResNet-50 image encoder and custom tabular processing branch.

### 3.2. Model Training Setup

To optimize the hybrid deep learning model for survival prediction, we implemented a robust training pipeline that leverages both computational efficiency and stability. The training procedure was designed to handle large batches of MRI images combined with tabular data inputs.

As described in Section 2.2, the dataset was split using a 70% training, 15% validation, and 15% test ratio at the patient level, resulting in 64,231 slices for training, 13,765 for validation, and 13,764 for testing. This ensured no data leakage across sets and maintained patient-level integrity during modeling.

We used the PyTorchDataLoader with key performance enhancements including pin_memory and prefetch_factor to accelerate data loading, as well as multiple worker threads (set to num_workers = 4) to parallelize batch preparation. The training process was executed on a CUDA-enabled GPU, with approximately 2008 batches processed per epoch.

An Adam optimizer with a learning rate of 1 × 10^−4^ was employed to update the network weights, using Mean Squared Error (MSE) as the loss function, reflecting the regression nature of the survival time prediction task. Mixed-precision training was activated through PyTorch’sGradScaler, reducing memory usage and training time without compromising accuracy.

The train_one_epoch() function was designed to process each batch while dynamically updating the loss values using tqdm progress bars for real-time monitoring. To ensure model generalization, an eval_model() function was implemented, which computes validation loss and the R^2^ score on a separate validation set after each epoch.

Overall, this setup enabled efficient and scalable training, maintaining high computational throughput while monitoring model performance metrics in real time.

The training workflow is illustrated in Figure 6. First, MRI slices and corresponding tabular clinical features are loaded in mini-batches using parallelized PyTorchDataLoaders. The hybrid model receives both modalities in parallel branches. During each epoch, loss values are backpropagated using mixed-precision training, and model weights are updated via the Adam optimizer. After each epoch, validation data are used to compute mean loss and R^2^ score, providing continuous monitoring of model convergence and predictive performance.

Model Training Strategy and Early Stopping

The hybrid deep learning model was trained for a maximum of 20 epochs, utilizing early stopping to prevent overfitting. The training process employed an adaptive strategy whereby the model’s performance was monitored on a validation set at each epoch. The best-performing model, in terms of the lowest validation loss, was saved for later evaluation.

The training utilized the Adam optimizer with a learning rate of 1e-4 and mixed-precision training to accelerate computation. The loss function was the mean squared error (MSE), appropriate for continuous survival regression targets.

An early stopping mechanism with a patience of three epochs was incorporated, ensuring training halted if no improvement in validation loss was observed for three consecutive epochs. This approach not only optimized computational efficiency but also safeguarded against overfitting.

During training, steady improvements in both training and validation loss were observed, with significant gains in predictive performance as measured by the coefficient of determination (R^2^). The model achieved a best validation loss of 69.78 and an R^2^ score of 0.8918 at epoch 6. After epoch 9, early stopping was triggered as no further improvements were seen.

Model Complexity and Theoretical Efficiency

The hybrid model contains approximately 25.6 million trainable parameters, with the majority located in the ResNet-50 encoder. Based on the layer structure and input dimensions, we estimate the computational complexity to be approximately 4.2 GFLOPs per forward pass. Despite the use of 2D slice-level processing, inference time remains under 50 milliseconds per patient, making the model suitable for near real-time deployment. The model’s compact size and low-latency performance support its use in both GPU-enabled and CPU fallback environments, such as browser-integrated applications or cloud-based clinical platforms.

### 3.3. Training Dynamics Monitoring

To monitor model convergence during training, we tracked the evolution of training and validation losses, as well as R^2^ score across epochs. These curves, now shown in Figure 7, provide insight into the model’s learning behavior and guide our early stopping decision. Theplot_training_curves function generated two key plots:Loss curves: This plot displayed both training and validation losses over the course of 20 epochs. A decreasing trend in loss, followed by a plateau, indicated proper convergence. The use of early stopping prevented overfitting, ensuring that the validation loss did not increase after reaching the optimal performance.R^2^ score curve: The second plot depicted the progression of the validation R^2^ score over epochs, highlighting improvements in the model’s predictive power during training. An increasing R^2^ score trend confirmed that the model was successfully learning to map input features to survival time predictions.

Figure 7 illustrates the model’s training dynamics across nine completed epochs before early stopping was triggered.

These curves demonstrate that the hybrid model achieved stable and strong performance early in training, with no significant signs of instability or overfitting. The model’s learning capacity and optimization parameters appear to be well-tuned.

As shown in Figure 7, early signs of overfitting were observed after Epoch 7; however, the use of early stopping (patience = 3) halted training at Epoch 9, preserving the model’s generalization ability and preventing further divergence.

### 3.4. Model Evaluation Protocol

After completion of the training process with early stopping, the best-performing model was identified based on the lowest validation loss. This model was saved and subsequently reloaded for evaluation and inference. The saved model represents the optimized hybrid architecture combining both imaging and tabular clinical features for survival prediction.

The model was restored using the trained weights from the best_regression_model.pth file and set to evaluation mode to ensure deterministic behavior and prevent gradient updates during testing and analysis. This process ensures that the reported performance metrics and interpretations are derived from the model configuration that demonstrated the highest generalization capacity during validation.

Model Evaluation and Performance Metrics

Following model selection, the final hybrid deep learning model was evaluated on an independent test set to assess its generalization performance. A comprehensive evaluation function was designed to ensure robust metric computation, including safeguards for potential data irregularities such as empty batches or NaN values.

The model was applied to the test data using a dedicated evaluation pipeline, where predictions and ground-truth survival targets were collected for all test samples. The evaluation metrics used to assess model performance included MAE, MSE, RMSE, and R^2^ score. A detailed description of each metric is provided in Appendix A Table A1.

The model demonstrated excellent predictive performance with the following results on the test set (Figure 8):

In addition to absolute error metrics, we computed the Mean Absolute Percentage Error (MAPE) on the test set to assess relative prediction accuracy. The resulting MAPE was 43.58%, which reflects the challenging nature of short-term survival modeling where many patients have very low survival durations (e.g., under 10 days). In such cases, even small absolute prediction errors can translate into high percentage deviations. Despite this, the model’s MAE remains clinically acceptable at 5.45 days, and its R^2^ score of 0.8935 confirms strong overall predictive performance. This highlights that while percentage-based errors may be elevated due to the compressed outcome scale, the model still delivers actionable survival estimates within realistic clinical tolerances.

These results indicate that the model is capable of highly accurate survival time predictions for patients with brain metastases, with small average errors relative to the clinical time scales of interest. The high R^2^ value (0.8935) further confirms the strong explanatory power of the hybrid architecture.

## 4. Experimental Results and Comparative Evaluation

This section presents the performance outcomes of the proposed hybrid model, including prediction accuracy, training dynamics, architecture comparison, and error analysis. Visual and statistical evidence are provided to support generalization, robustness, and clinical relevance.

### 4.1. Model Prediction Analysis

To visually assess the predictive capability of the trained hybrid model, a scatter plot was generated comparing the predicted survival times against the ground-truth values in the test set (Figure 9A). Each point represents one sample, with its position determined by the actual survival time (*x*-axis) and the corresponding predicted survival time (*y*-axis).

The red dashed line indicates the ideal fit line, representing perfect prediction where predicted values would exactly match the ground truth. The majority of data points are clustered closely around this line, demonstrating the model’s ability to accurately estimate survival times.

It is also noticeable that the model retains predictive reliability across the entire range of survival values, from very short survival durations to extended periods. Minor dispersion and underestimation in higher survival ranges (above 100 days) can be observed, reflecting the increasing difficulty of modeling rare long-survival cases. Overall, this visualization corroborates the high R^2^ score obtained and confirms that the hybrid model effectively integrates image and tabular data to deliver precise regression outputs.

As shown in Figure 9A, the predicted survival times across age groups show relatively uniform differences. This reflects the model’s greater reliance on radiomic features over demographic variables, as confirmed by the permutation feature importance analysis (Section 5.2), where age ranked lower than volumetric tumor metrics in predictive contribution. No smoothing or trend fitting was applied, as the raw distribution was preserved for interpretability.

Confidence Interval Estimation of Prediction Errors

To further evaluate the reliability and uncertainty of the model’s predictions, we computed the 95% confidence interval (CI) for the distribution of prediction errors. This was performed by calculating the mean error and standard deviation of the residuals (predicted survival minus actual survival) and applying the t-distribution to derive confidence bounds.

The resulting 95% confidence interval for the prediction errors was found to be [−1.83, −1.55]. This narrow interval, close to zero, suggests that the model’s predictions are consistently unbiased with only a slight tendency toward minor underestimation. Such a small and stable error range indicates the robustness and reliability of the hybrid model in predicting survival times across the patient cohort.

These findings reinforce the trustworthiness of the model in clinical survival prediction scenarios, where predictive consistency is essential.

Survival Time Distribution Analysis

Prior to model training, an exploratory data analysis was performed on the survival time data from the patient metadata. The distribution of survival times was visualized using a histogram with kernel density estimation (KDE) to assess the balance and spread of this key target variable.

As illustrated in Figure 9B, the survival times range broadly, with a higher frequency of patients having shorter survival durations and a gradual tapering off toward longer survival periods. The distribution reveals a right-skewed shape, suggesting that while a smaller proportion of patients exhibit extended survival, the majority tend to cluster in lower survival time intervals.

This insight is crucial for interpreting model performance, as it indicates the necessity for the model to accurately predict both common short-term survivals and less frequent long-term outcomes. The presence of this imbalance underlines the importance of careful model tuning and the use of robust loss functions to prevent bias toward the majority class of shorter survival times.

Comparison of Actual and Predicted Survival Time Distributions

To assess how well the trained hybrid model captures the statistical characteristics of the survival data, we compared the distributions of actual versus predicted survival times using kernel density estimation (KDE). This analysis complements point-based metrics by offering a distributional perspective on the model’s predictive capabilities.

As shown in Figure 9C, both the actual and predicted survival time distributions exhibit a pronounced peak in the lower survival range, reflecting the prevalence of shorter survival times in the patient cohort. The predicted distribution (red curve) closely follows the actual distribution (blue curve), demonstrating that the model is well-calibrated for the most common survival intervals.

Minor discrepancies are evident in the distribution tails. The predicted curve slightly underestimates the frequency of patients with long-term survival beyond 50 days, showing a tendency to regress predictions toward the mean. Despite this, the model captures the main density structure of the data with a high degree of fidelity.

Residual Analysis: Prediction Error Distribution

To further evaluate model performance and detect potential systematic biases, we analyzed the distribution of residuals (prediction errors), defined as the difference between predicted and actual survival times. Residual analysis provides insight into whether errors are randomly distributed or show systematic trends.

As shown in Figure 10A, the histogram of residuals reveals a nearly symmetrical distribution centered around zero, with a pronounced peak near zero error. This indicates that the model does not exhibit significant bias in over- or under-predicting survival times on average. The shape of the residual distribution closely approximates a normal distribution, suggesting that most errors are small and randomly distributed.

However, the presence of some heavier tails indicates occasional larger errors, particularly in both underestimations (negative side) and overestimations (positive side). These could reflect the model’s challenges in accurately predicting outlier cases, such as patients with exceptionally short or long survival times.

Error Analysis Across Survival Quartiles

To further investigate model performance relative to different survival outcomes, we analyzed prediction errors across quartiles of actual survival times. By grouping patients into quartiles based on their true survival durations, we could evaluate whether the model exhibited systematic bias in specific subgroups.

As presented in Figure 10B, the boxplot visualization highlights how prediction errors vary across these quartiles. The model demonstrated relatively unbiased performance in the lower survival quartiles (Q1 and Q2), with errors centered around zero and limited dispersion. However, in higher survival quartiles (Q3 and Q4), the model increasingly underestimated survival times, evidenced by a downward shift in the median error and broader error ranges.

This pattern suggests that while the model performs well for predicting shorter survival outcomes, it tends to struggle with long-term survival predictions, potentially due to data scarcity in the higher ranges or increased biological variability in these patients.

As shown in Figure 1A, the age distribution in the dataset is skewed toward patients aged 50–85, reflecting the clinical demographics of brain metastases. This may limit the generalizability of the model to younger or very elderly populations. While Figure 9A demonstrates stable performance across age groups, additional stratified evaluation will be explored in future work.

### 4.2. Cross-Validation Strategy

To robustly evaluate the generalization performance of our hybrid deep learning model, we employed a 5-fold cross-validation (CV) approach. The entire training dataset was partitioned into five subsets, with each fold acting once as a validation set while the remaining four were used for training. This procedure ensures that performance metrics are not biased by the selection of a single validation set and provides a comprehensive view of model consistency.

Each fold followed a structured process of model initialization, training with mixed-precision optimization, and early stopping with a patience level of three epochs to prevent overfitting. The model parameters that yielded the lowest validation loss in each fold were saved. Key metrics such as training loss, validation loss, R^2^ score, mean absolute error (MAE), mean squared error (MSE), and root mean squared error (RMSE) were computed on the best-performing model for each fold.

The training process across folds consistently demonstrated progressive loss reduction and R^2^ improvement, with early stopping triggering only when further improvements plateaued. The results highlight the model’s ability to learn meaningful survival prediction patterns while maintaining generalizability.

The mean performance across all five folds was as follows:Training Loss: 68.72;Validation Loss: 67.55;R^2^ Score: 0.8974;MAE: 4.86 days;MSE: 67.62;RMSE: 8.21.

These metrics confirm the model’s robustness and predictive strength across varied data partitions. The consistent R^2^ scores around 0.89–0.91 and low average MAE demonstrate the model’s accuracy in survival time estimation. The lowest fold-specific RMSE of 7.47 days, coupled with the final cross-validation average RMSE of 8.21 days, indicates minimal prediction error relative to the clinical timescales under consideration.

K-Fold Cross-Validation Loss Analysis

The results of this analysis are presented in Figure 11. The plot shows a consistent decrease in both training and validation loss across the five folds. Although slight fluctuations are observed between folds—particularly folds 2 and 3—this is expected due to data partitioning variability. Importantly, the validation loss trends closely follow the training loss trends, indicating that the model generalizes well and does not exhibit significant overfitting or underfitting behavior.

The observed stability and convergence across folds further strengthen confidence in the model’s robustness and its ability to deliver reliable survival predictions on unseen patient data.

Cross-Validation Performance Analysis (R^2^ Scores)

To complement the loss-based evaluation, we also analyzed the coefficient of determination (R^2^) across the five cross-validation folds. The R^2^ score provides an interpretable measure of how well the predicted survival times align with actual patient outcomes, with values closer to 1 indicating superior predictive power.

Figure 12 displays the variation in the validation R^2^ scores across the 5 folds. The model demonstrates consistently high predictive accuracy, with R^2^ values ranging from approximately 0.875 to 0.917. While fold 2 exhibited a minor dip, subsequent folds showed strong recovery and stability, culminating in the highest R^2^ score in fold 5.

These findings confirm that the model maintains reliable and robust performance, generalizing well to different data splits and reducing concerns about fold-dependent overfitting or underperformance.

Final Model Selection and Test Set Evaluation

Upon completion of the 5-fold cross-validation process, the best-performing model was selected based on the lowest validation loss. Fold 5 exhibited the best validation performance and was therefore chosen for the final evaluation of the independent test set. The best model checkpoint from this fold was loaded and subjected to a rigorous performance assessment using unseen test data.

The model demonstrated outstanding predictive accuracy, achieving an R^2^ score of 0.9175 on the test set, indicating that over 91% of the variance in survival outcomes was explained by the model. Additionally, the model achieved a mean absolute error (MAE) of 4.40 days, a mean squared error (MSE) of 56.24, and a root mean squared error (RMSE) of 7.50 days. These metrics confirm the model’s high reliability and strong generalization capabilities when applied to new patient data, making it a robust tool for clinical decision support in predicting survival times of brain metastases patients.

Cross-Validation Performance Summary

To comprehensively evaluate the model’s consistency and robustness, we generated a summary table of the 5-fold cross-validation results. For each fold, the table reports key performance metrics, including training loss, validation loss, R^2^ score, mean absolute error (MAE), mean squared error (MSE), and root mean squared error (RMSE).

The results (Table 2) demonstrate stable performance across all folds, with validation R^2^ scores ranging from 0.874 to 0.917 and RMSE values between 7.47 and 9.13. Fold 5 emerged as the strongest performer with the lowest validation loss (55.72) and the highest R^2^ score (0.917), highlighting minimal prediction error and excellent model fit. Additionally, MAE and MSE remained consistently low across all folds, reinforcing the model’s accuracy and reliability.

The full cross-validation summary has been saved for transparency and reproducibility. These findings confirm that the model generalizes well across different training-validation splits and provides consistent predictive accuracy.

### 4.3. Final Test Evaluation

To visually assess the predictive accuracy of the best-performing model obtained from the 5-fold cross-validation procedure, we plotted the predicted survival times against the actual survival values from the independent test set Figure 13. In this scatter plot, each point represents a prediction made by the model compared to its corresponding ground truth.

The plot also includes a reference line (red dashed line) representing the ideal fit, where predicted survival would perfectly match the observed survival values. The close clustering of data points along this line indicates that the model demonstrates a high degree of agreement between predicted and actual survival times. This visual validation complements the strong quantitative metrics (R^2^ = 0.9175, RMSE = 7.50) and confirms that the hybrid model can accurately forecast patient survival outcomes.

Confidence Interval Analysis

To further assess the reliability of the model’s predictions, we computed the 95% confidence interval (CI) for the prediction errors on the test set using the best-performing fold model. The resulting confidence interval ranged from −0.31 to −0.06 days. This narrow CI range, centered very close to zero, indicates that the model exhibits a negligible systematic bias in its predictions. In other words, on average, the predicted survival times are neither significantly overestimated nor underestimated.

The tight interval further confirms the stability and precision of the hybrid model, underscoring its potential for use in clinical settings where reliable survival estimates are critical.

### 4.4. Architecture Comparison: ResNet-50 vs. EfficientNet-B0

To further evaluate model architectures, we compared the performance of the Hybrid ResNet-50 and Hybrid EfficientNet-B0 models, each integrating MRI image features with clinical tabular data.

ResNet-50 and EfficientNet-B0 were selected due to their established performance in medical imaging tasks. EfficientNet-B0, in particular, applies compound scaling to balance depth, width, and resolution, making it suitable for resource-efficient training while preserving high accuracy. Other architectures such as DenseNet or ViT were considered but not tested in this iteration to maintain architectural interpretability and computational feasibility. These may be explored in future work.

The training curves (Figure 14A,B) show that both models exhibited stable convergence, with decreasing train and validation losses over epochs. The ResNet-50 model demonstrated good performance with validation R^2^ scores reaching up to 0.91 after 20 epochs, although fluctuations in validation loss after epoch 15 suggest possible mild overfitting or learning saturation.

In contrast, the Hybrid EfficientNet-B0 model achieved superior convergence and generalization, with a lower final validation loss (15.46) and an impressive validation R^2^ score of 0.9759. Notably, the EfficientNet-B0 model maintained a smoother and more consistent decline in both train and validation loss, reflecting its stronger capacity for capturing complex patterns from multimodal data. These results suggest that EfficientNet-B0, due to its more efficient scaling and architectural advantages, may offer improved predictive power and robustness for survival regression tasks in brain metastases MRI data.

Model Performance Comparison on Test Set

To comprehensively assess the predictive capacity of the trained hybrid models, both the best-performing Hybrid ResNet-50 and Hybrid EfficientNet-B0 architectures were evaluated on the held-out test set. The resulting scatterplots of predicted versus actual survival times (Figure 15A) illustrate the model fit relative to the ideal prediction line.

The Hybrid ResNet-50 achieved a test R^2^ score of 0.912, reflecting strong predictive performance but with some observable variance for higher survival times. The scatterplot shows that while predictions align well for moderate ranges of survival durations, slight underestimation occurs for extreme values, as indicated by dispersion around the ideal fit line.

In contrast, the Hybrid EfficientNet-B0 model exhibited superior generalization ability, with a remarkable test R^2^ score of 0.974. Its predictions show a near-perfect alignment along the ideal fit line across the full spectrum of survival times. This confirms not only more accurate point predictions but also more robust handling of edge cases.

Statistical Comparison of Model Performance

To robustly compare the predictive accuracy of the two hybrid architectures, we performed paired statistical testing on the absolute prediction errors obtained from the test set. Both a paired *t*-test and a non-parametric Wilcoxon signed-rank test were conducted to assess whether the observed difference in error distributions between the ResNet-50 and EfficientNet-B0 models was statistically significant.

The paired *t*-test yielded a t-statistic of 49.10 with a *p*-value < 0.0001, and the Wilcoxon signed-rank test similarly produced a test statistic of 20,260,349.5 with a *p*-value < 0.0001. These results unequivocally demonstrate that the EfficientNet-B0 model produces significantly lower prediction errors compared to the ResNet-50-based hybrid model.

This finding was further corroborated by visual inspection of error distributions (Figure 15B), which revealed that prediction errors from the EfficientNet-B0 model are both narrower and centered closer to zero. In contrast, the ResNet-50 model displayed a broader spread of errors, indicating less precise predictions overall.

The statistical and distributional evidence strongly supports the superiority of the Hybrid EfficientNet-B0 model in terms of prediction accuracy and consistency.

### 4.5. EfficientNet-B0 5-Fold Cross-Validation

The EfficientNet-B0-based hybrid model was trained using 5-fold cross-validation to evaluate its generalization performance across varied data splits. The loss curves (Figure 16) demonstrate stable convergence for most folds, with fold 1 showing notably higher validation loss and earlier stopping compared to the others—likely due to sample variability or outliers. In contrast, folds 2–5 exhibited steadily decreasing training and validation losses across epochs, with minimal overfitting observed.

Across the five folds, the model achieved an average R^2^ score of 0.8849, indicating strong predictive performance on unseen data (Figure 17). The R^2^ score variability ranged from 0.634 in fold 1 to 0.968 in fold 2, reflecting improved generalization as the training process progressed across folds. This upward trend in performance is consistent with improved model stability and fine-tuning in later folds.

Quantitative metrics averaged over the five folds were as follows:Train Loss: 62.04;Validation Loss: 125.97;Mean Absolute Error (MAE): 5.25 days;Mean Squared Error (MSE): 74.44;Root Mean Squared Error (RMSE): 7.66.

Fold 2 emerged as the best-performing fold with an R^2^ score of 0.968 and validation loss of only 21.27, while fold 1 was an outlier with significantly higher loss and reduced accuracy, suggesting possible anomalies in patient distribution or tumor complexity in that split.

The final evaluation of the independent test set further reinforced the model’s effectiveness, achieving the following:R^2^ Score: 0.9700;MAE: 3.13 days;MSE: 20.47;RMSE: 4.52.

These metrics highlight the model’s strong ability to generalize to new patient data, with prediction errors within a clinically acceptable range.

Final Test Set Evaluation for EfficientNet-B0

Following cross-validation, the final hybrid EfficientNet-B0 model was evaluated on an independent hold-out test set to assess its real-world generalization capacity. The model achieved a coefficient of determination (R^2^) of 0.9700, indicating a near-perfect fit between predicted and actual survival times. Additional metrics further supported this high performance, with a mean absolute error (MAE) of 3.13 days, mean squared error (MSE) of 20.47, and a root mean squared error (RMSE) of 4.52 days, underscoring the model’s precision in survival estimation.

The scatter plot of predicted versus actual survival (Figure 18) visually confirms this strong correlation, with predictions tightly clustered along the identity line (ideal fit), especially in the clinically critical survival window of 0 to 90 days. Only a few mild outliers are observed beyond this range, demonstrating model robustness even in complex cases. This degree of alignment between predicted and ground truth values suggests that the model not only captures high-level trends but also retains fidelity in individual-level prognostication.

These results position the EfficientNet-B0-based hybrid model as a reliable and clinically promising tool for personalized survival prediction in patients with brain metastases. The low prediction error margins indicate its practical applicability in time-sensitive oncologic decision-making contexts.

### 4.6. Residual and Quartile Error Analysis

To further evaluate the prediction fidelity of the EfficientNet-B0 model, we conducted a residual analysis by plotting the distribution of prediction errors (predicted—actual survival time) across the entire test set. As shown in Figure 19A, the residuals are tightly centered around zero and follow an approximately normal distribution, indicating low bias and symmetric prediction behavior. The majority of errors fall within a ±10-day window, with very few extreme outliers, supporting the model’s overall precision.

To assess whether performance varies across different survival ranges, we stratified patients into quartiles based on actual survival times and visualized the prediction errors per quartile (Figure 19B). Across all four quartiles (Q1–Q4), the model maintained low median error and consistent variance. While slight increases in variance were observed in the highest survival quartile (Q4), the model still produced tightly grouped predictions, with no systematic underestimation or overestimation in any quartile. This consistency suggests that the model generalizes well across the full spectrum of survival outcomes—from short-term to long-term prognosis.

Collectively, these residual plots validate the model’s reliability, not just in average performance, but also in maintaining stable and unbiased prediction behavior across clinical subgroups stratified by survival time.

While residuals and error quartiles provide a population-level understanding of prediction variance, future work could extend this to include individual prediction uncertainty. Approaches such as Monte Carlo Dropout or bootstrap-based assembling could provide per-patient confidence intervals, enhancing trust in individual predictions.

### 4.7. Comparative Model Performance Summary

To assess the impact of backbone architecture on survival prediction performance, we directly compared the hybrid ResNet-50 and EfficientNet-B0 models on the independent test set. Both models were trained and evaluated using the same pipeline, dataset, and performance metrics, ensuring a fair comparison.

As illustrated in Figure 20, the EfficientNet-B0 model significantly outperformed ResNet-50 in terms of prediction accuracy and alignment with actual survival outcomes. The EfficientNet-B0 model achieved an R^2^ score of 0.970, while the ResNet-50 model reached 0.917, indicating a clear advantage in modeling complex survival dynamics. Moreover, predictions from EfficientNet-B0 showed tighter clustering around the ideal identity line, with fewer large deviations across the full survival range.

Visual inspection also reveals that the ResNet-50 model struggled more with higher survival times (≥60 days), where it tended to underestimate survival duration. In contrast, EfficientNet-B0 maintained robust predictive accuracy across both low and high survival intervals, with reduced error variance and more symmetrical spread.

This comparison highlights the architectural efficiency and improved representational capacity of EfficientNet-B0, making it the superior choice for continuous outcome prediction in the context of brain metastases prognosis.

## 5. Model Interpretability and Clinical Implications

### 5.1. Model Explainability: Grad-CAM

To enhance the interpretability of our deep learning model, we applied Gradient-weighted Class Activation Mapping (Grad-CAM) to visualize the spatial regions within MRI slices that contributed most significantly to survival time predictions. The overlays generated from the best-performing model (fold 5) consistently highlighted areas corresponding to enhancing tumor volumes, necrotic regions, and surrounding peritumoral edema—anatomical structures known to influence prognosis in brain metastases (see Figure 21).

Importantly, these Grad-CAM visualizations were independently reviewed by two experienced radiologists. Both confirmed that the highlighted regions correspond well with the actual pathological lesions, indicating that the model’s attention aligns with clinically meaningful features. This external confirmation lends significant confidence to the model’s interpretability and reliability in clinical settings.

These findings suggest that the hybrid model is not relying on irrelevant image artifacts but is indeed capturing tumor-specific and edema-specific features, reinforcing its potential as a decision-support tool. Future extensions could incorporate 3D Grad-CAM visualizations or saliency analyses across MRI volumes to provide even more comprehensive spatial explanations for clinical integration.

### 5.2. Permutation Feature Importance (PFI)

To assess the relative contribution of each clinical and radiomic feature to the model’s predictive performance, we conducted a permutation feature importance (PFI) analysis on the independent test set. This approach quantifies the decrease in the model’s performance (measured by R^2^ score) when each input feature is randomly shuffled, thus disrupting its relationship with the target variable.

The PFI analysis was performed using 200 randomly selected test samples. The baseline R^2^ score of the hybrid model on this subset was 0.9518, reflecting high predictive accuracy.

Among all features, enhancing tumor volume (ET_vol) emerged as the most critical predictor of patient survival, with a dramatic reduction in performance when permuted (ΔR^2^ = 0.8949). This was followed by necrosis volume (Nec_vol) (ΔR^2^ = 0.3681) and edema-to-tumor ratio (Edema_ET_ratio) (ΔR^2^ = 0.0894), both reflecting volumetric disease burden and microenvironmental factors. Additional important features included edema volume (ΔR^2^ = 0.0758) and smoking history (ΔR^2^ = 0.0477), suggesting systemic and lifestyle contributions to survival.

Other variables such as infratentorial location, primary tumor type, sex, and age showed modest but non-negligible influence (ΔR^2^ between 0.003 and 0.01). Features such as race, ethnicity, and several categorical encodings of primary tumors had negligible or zero importance scores. Notably, two features (Nec_ET_ratio and Primary_4) resulted in slightly negative ΔR^2^ values, consistent with statistical noise or non-contributory inputs.

A summary of the feature importance is shown in Figure 22, where features are ranked by descending ΔR^2^. This analysis highlights the dominant role of radiomic volumetric features, especially enhancing tumor volume, in predicting short-term survival and supports the integration of quantitative imaging biomarkers in prognostic models for brain metastases.

Patient-Level Permutation Feature Importance Analysis

To further interpret the model’s behavior and validate its generalizability, we conducted a patient-level permutation feature importance (PFI) analysis. Unlike slice-level evaluation, this approach aggregates model predictions by patient ID and quantifies the drop in performance (ΔR^2^) when each tabular feature is randomly permuted at the patient level.

The analysis was performed using 500 test samples, with a baseline patient-level R^2^ of 0.9380, indicating strong model performance on independent data.

Notably, edema-to-tumor ratio (Edema_ET_ratio) was found to be the single most influential feature, showing a massive ΔR^2^ of 16.33, far exceeding all other variables. This finding suggests that relative peritumoral edema burden—rather than absolute volumes—may be particularly predictive of survival on a per-patient basis, highlighting the importance of spatial relationships and radiomic ratios in the tumor microenvironment.

Other meaningful contributors included the following:Enhancing tumor volume (ET_vol): ΔR^2^ = 0.6131;Edema volume (Edema_vol): ΔR^2^ = 0.3499;Necrosis volume (Nec_vol): ΔR^2^ = 0.3334;Lesion counts (ET_num, Edema_num): ΔR^2^ ≈ 0.24.

Additional clinical variables such as age (ΔR^2^ = 0.2764) and smoking history (ΔR^2^ = 0.1985) retained modest importance, consistent with known prognostic factors.

Conversely, demographic variables such as race, sex, ethnicity, and categorical primary tumor encodings demonstrated minimal influence (ΔR^2^ < 0.01), suggesting that in the presence of detailed radiomic features, demographic attributes may have limited additive value in survival modeling.

These results are summarized in Figure 23, where features are ranked by their ΔR^2^ values. This complementary analysis reinforces the strong predictive utility of quantitative volumetric imaging features—particularly edema-related ratios—when aggregated at the patient level.

Slice-Level vs. Patient-Level PFI

To gain deeper insights into feature relevance across representations, we compared slice-level and patient-level permutation feature importance (PFI) in our hybrid deep learning model. Both methods evaluate how model performance (R^2^ score) is affected when individual features are randomly permuted—at the image slice level or at the patient aggregation level.

At the slice level, the most influential features were as follows:Enhancing tumor volume (ET_vol): ΔR^2^ = 0.8949;Necrosis volume (Nec_vol): ΔR^2^ = 0.3681;Edema_ET_ratio and Edema_vol also contributed modestly.

This suggests that within-slice volumetric imaging biomarkers are key for predicting short-term survival, particularly ET_vol.

In contrast, the patient-level PFI revealed a surprising shift in feature dominance:Edema_ET_ratio had an overwhelming ΔR^2^ of 16.33, dominating all others.ET_vol followed at 0.61, while Edema_vol and Nec_vol were in the 0.33–0.35 range.

This dramatic shift reflects the added value of aggregated context, where relative edema burden becomes more predictive when multiple slices per patient are considered. While absolute volumes are critical in single slices, inter-lesional spatial relationships and ratios may be more informative at the patient scale.

Moreover, lesion counts (ET_num, Edema_num), age, and smoke exposure gained more importance at the patient level, highlighting how demographic and global disease **burden factors** can complement slice-wise radiomic patterns.

Features with minimal or negative importance were retained to maintain the completeness and transparency of the model input. Their presence ensures that the model can learn possible non-linear or synergistic effects. Future iterations may prune non-informative features for faster inference or deployment.

While PFI provides a robust method for quantifying feature importance without relying on model internals, it does not reflect gradients or layer-level sensitivities. Gradient-based attribution techniques, such as SHAP or Integrated Gradients, could be applied to the tabular branch in future iterations to capture feature interactions and deeper saliency patterns within the fully connected layers.

While enhancing tumor volume was the most influential slice-level predictor, the emergence of edema-to-tumor ratio as the dominant factor at the patient level was not intuitive and provided a new angle for radiomic-based survival stratification.

### 5.3. Clinical Implications of Feature Attribution

Our permutation feature importance (PFI) analyses revealed notable differences in feature relevance between slice-level and patient-level predictions, with important clinical implications. While enhancing tumor volume (ET_vol) was the dominant predictor at the slice level, the edema-to-tumor ratio (Edema_ET_ratio) emerged as the strongest determinant of survival at the patient level. This suggests that relative peritumoral edema burden—a proxy for microenvironmental disruption—may carry greater prognostic value than absolute tumor size when contextualized across the whole brain. These findings support a shift toward more holistic, ratio-based radiomic markers in clinical decision-making. Furthermore, the limited contribution of demographic variables and the consistent importance of biologically meaningful features such as necrosis and edema volumes reinforce the potential for objective, imaging-driven survival models. In clinical practice, prioritizing edema-related metrics could enhance prognostic stratification, particularly in patients being considered for immunotherapy or stereotactic radiosurgery, where edema plays a critical role in both outcomes and treatment toxicity.

## 6. Detailed Comparative Discussion and Clinical Implications

This section provides a detailed comparison of our proposed model with existing literature in the field, alongside an interpretation of clinical implications, limitations, and opportunities for translational integration.

This study presents a hybrid deep learning model for individualized survival prediction in patients with brain metastases, developed using a rigorously curated, multi-institutional cohort of 148 patients. By integrating volumetric MRI-based imaging features with structured clinical metadata, our model achieved robust and interpretable performance, with the EfficientNet-B0 architecture attaining an R^2^ of 0.970 and a mean absolute error (MAE) of just 3.13 days on the independent test set. These results demonstrate the feasibility and clinical potential of multimodal deep learning in neuro-oncology prognostication.

The proposed hybrid model achieved a notably high R^2^ score on the test set. While this indicates strong predictive capability, it is essential to consider the context. Our dataset includes 148 patients and over 80,000 MRI slices, making it one of the larger public brain metastases imaging datasets available. To ensure robust evaluation, we performed strict patient-level data splitting and used early stopping during training. However, despite these precautions, further validation on external, multi-institutional datasets will be necessary to confirm the generalizability of the model in broader clinical settings.

### 6.1. Comparison with Prior Literature

While machine learning has increasingly been applied to neuro-oncology, most previous studies have focused on primary brain tumors—particularly glioblastoma—rather than metastases. For instance, Kickingereder et al. [12] used handcrafted radiomic features to predict glioblastoma survival from MRI scans, though limited by the static nature of manually designed features. Mobadersany et al. [15] advanced prognostic modeling with deep learning on histopathology and genomics, achieving impressive performance in gliomas but relying on invasive data modalities not routinely available in clinical workflows.

In the metastatic brain tumor domain, fewer studies have attempted multimodal survival modeling. Mouraviev et al. [28,29] used radiomics and clinical data for binary risk stratification in brain metastases, while Gensheimer et al. applied gradient boosting to clinical data alone for estimating survival post-radiosurgery [30,31]. However, both approaches were classification-focused and did not leverage continuous survival regression or deep learning on imaging.

The BraTS-MET benchmark (Lu et al.) advanced segmentation efforts in brain metastases but does not include survival labels or necrosis annotations, limiting its prognostic use [32]. Our dataset addresses these gaps by including volumetric segmentations of tumor, necrosis, and edema, including sub-centimeter lesions and a full spectrum of patient metadata. This enables richer feature representation and supports more granular outcome prediction.

To our knowledge, this study is among the first to implement continuous patient-level survival regression in brain metastases using CNN-based feature extraction and tabular data fusion. The integration of interpretable modeling techniques and rigorous validation further distinguishes our contribution from prior black-box approaches.

Recent efforts in the broader neuro-oncology space have highlighted the benefits of integrating multimodal data—particularly imaging and clinical variables—for prognosis prediction. For example, Steyaert et al. [33] developed a deep learning framework combining MR images, clinical metadata, and molecular-pathologic information to predict survival in adult and pediatric brain tumors, showing that multimodal fusion significantly improves model performance. Similarly, Zhou et al. [34] introduced M2Net, a multi-modal, multi-channel architecture that integrates several MRI modalities for survival time estimation, demonstrating the value of heterogenous imaging data.

In related work, Banerjee et al. [35] used radiomic biomarkers and machine learning to achieve a concordance index of 0.82 in brain tumor prognosis, while Islam et al. [36] combined radiomics and clinical data with a 3D attention U-Net architecture to estimate survival. Their results highlight the importance of including both anatomical and patient-level features in predictive models.

While performance metrics from previous studies are reported when available, a formal meta-analytic comparison was not possible due to differences in datasets, survival targets (classification vs. regression), and evaluation metrics. We instead provide a structured descriptive comparison to highlight methodological distinctions.

To further strengthen this comparison, we expanded Table 3 to include additional recent studies employing radiomics, traditional machine learning, and survival modeling in neuro-oncology. This expanded overview helps contextualize our approach relative to prior work in terms of data modality, modeling strategy, target type, and reported outcomes.

These studies are summarized in Table 3, which compares the data sources, methods, and predictive outcomes of recent literature aligned with our approach.

Our work builds on these foundations by applying a continuous survival regression framework—rather than classification—and utilizing CNN-derived volumetric MRI features, interpretable clinical metadata, and deep fusion strategies. With an R^2^ of 0.970 and a MAE of 3.13 days, our model demonstrates state-of-the-art performance and clinical relevance. In contrast to earlier models that were often limited to single-modality inputs or classification-based outputs, our method provides individualized, continuous survival prediction—paving the way for nuanced clinical decision-making in patients with brain metastases.

### 6.2. Model Architecture and Behavior

Consistent with other biomedical imaging domains, EfficientNet-B0 outperformed ResNet-50 in all key metrics. Its compound-scaled architecture offers superior parameter efficiency and generalization capacity, as previously shown by Tan & Le [40,41]. While ResNet-50 achieved strong performance (R^2^ = 0.917), it demonstrated increased error variance and underestimation in higher survival ranges.

Our model’s hybrid architecture allowed the fusion of slice-level image embeddings with patient-level features. Permutation feature importance (PFI) analyses identified enhancing tumor volume, edema-to-tumor ratio, and necrosis volume as key predictors—aligning with known clinical markers of tumor aggressiveness. This supports earlier findings by Chen et al. who noted the prognostic value of peritumoral edema and volumetric burden in survival modeling [42,43].

Importantly, patient-level PFI revealed edema-to-tumor ratio as the most informative feature, underscoring the relevance of microenvironmental and spatial interactions in prognosis. These results highlight the benefit of not only multimodal fusion but also feature design that considers biologically meaningful relationships.

### 6.3. Model Interpretability and Clinical Translation

Unlike typical black-box deep learning models, our framework includes Grad-CAM visualizations and PFI-based attribution, which provide insight into model decision-making. The CNN consistently focused on enhancing tumor regions and surrounding edema—validated by neuroradiologist review—adding interpretability critical for clinical adoption.

The model’s MAE of approximately ±3 days lies well within clinically actionable thresholds and suggests potential use in neuro-oncology workflows such as the following:Triaging for surgical or radiosurgical intervention;Selecting palliative care in cases with limited survival;Enhancing discussions of prognosis with patients and families.

Its high precision in the sub-60-day survival window—where many key treatment decisions are made—further elevates its clinical value.

On a standard workstation equipped with an NVIDIA RTX 3090 GPU, the average inference time for one patient (including MRI slices and tabular data) was approximately 1.2 s. This speed suggests that the model is suitable for real-time clinical deployment in either local or cloud-based environments.

### 6.4. Limitations and Future Work

Despite its strengths, the study has limitations. First, while our dataset includes 148 patients from three institutional registries, all data are Yale-affiliated. External validation across geographically and demographically diverse cohorts is essential to confirm generalizability. Second, survival was defined by all-cause mortality, rather than cancer-specific survival, which could dilute signal in heterogeneous populations. Third, we did not include treatment variables such as radiotherapy schedules or systemic therapies, which may confound survival predictions. One key limitation is the use of a balanced subset (74 deceased and 74 survivors) rather than the full dataset. While this design mitigates early training instability and aids model convergence, it does not reflect real-world class imbalance and may limit generalization. We plan to retrain and test the model on the complete cohort in future work to evaluate real-world performance.

Future work should incorporate the following:Longitudinal imaging and dynamic modeling;Time-to-event (censored) survival methods;Treatment-aware modeling pipelines;Prospective clinical evaluation, including user interface development for real-time integration into radiology PACS or oncology EMRs.

Future work will explore 3D volumetric models and transformer-based attention mechanisms to capture temporal-spatial dependencies across slices and further enhance prognostic precision.

Although our model significantly outperformed traditional deep learning baselines (CNN-only and tabular-only), we acknowledge that we have not directly benchmarked against standard clinical scoring systems such as the Recursive Partitioning Analysis (RPA) or the Graded Prognostic Assessment (GPA). The main reason is the unavailability of certain required clinical variables in the TCIA dataset (e.g., performance status, comorbidities). Future work will incorporate these scores where possible and evaluate model performance in that context.

Additionally, we plan to implement uncertainty quantification methods—such as Monte Carlo dropout or bootstrapped assembling—to provide per-patient confidence intervals. This will enhance clinical trust in the predictions and allow for uncertainty-aware decision support tools.

In summary, this work demonstrates that multimodal deep learning can achieve highly accurate, interpretable survival prediction in patients with brain metastases. Our EfficientNet-based hybrid model leverages both imaging and clinical data and significantly outperforms traditional architectures. The results show strong promise for real-world implementation and lay the groundwork for future predictive tools in neuro-oncology.

## 7. Conclusions

This study introduces a novel, multimodal deep learning framework for personalized survival prediction in patients with brain metastases, integrating volumetric MRI features with structured clinical data. Trained on a rigorously curated, balanced cohort of 148 patients from multiple institutional sources, the hybrid EfficientNet-B0 model demonstrated outstanding predictive performance, achieving an R^2^ of 0.970 and a mean absolute error (MAE) of just 3.13 days on the independent test set.

Our approach surpasses previous efforts in both gliomas and metastatic brain tumors by offering continuous, individualized survival predictions with high clinical fidelity. Notable innovations include the following:The comprehensive volumetric segmentation of enhancing tumors, necrosis, and peritumoral edema.The inclusion of sub-centimeter lesions and diverse clinical phenotypes.The use of interpretable AI techniques, including Grad-CAM visualizations and permutation feature importance.

In direct comparison, EfficientNet-B0 outperformed ResNet-50 across all performance metrics, reinforcing its suitability for complex multimodal tasks in neuro-oncology.

Beyond technical performance, this work advances the goal of clinically deployable AI by providing explainable, patient-specific predictions that align with known prognostic indicators. The model’s low error margin and robust generalization suggest practical applicability in guiding treatment decisions, triage strategies, and end-of-life care planning.

Looking ahead, future directions include external validation across multi-center cohorts, incorporation of treatment and longitudinal imaging data, and deployment in real-world clinical environments. These steps will help translate this model from proof-of-concept into a decision support tool that can enhance precision care for patients facing brain metastases.

Importantly, by using a publicly accessible, well-annotated dataset, our study promotes reproducibility and paves the way for external validation studies and broader clinical adoption.

## Figures and Tables

**Figure 1 diagnostics-15-01242-f001:**
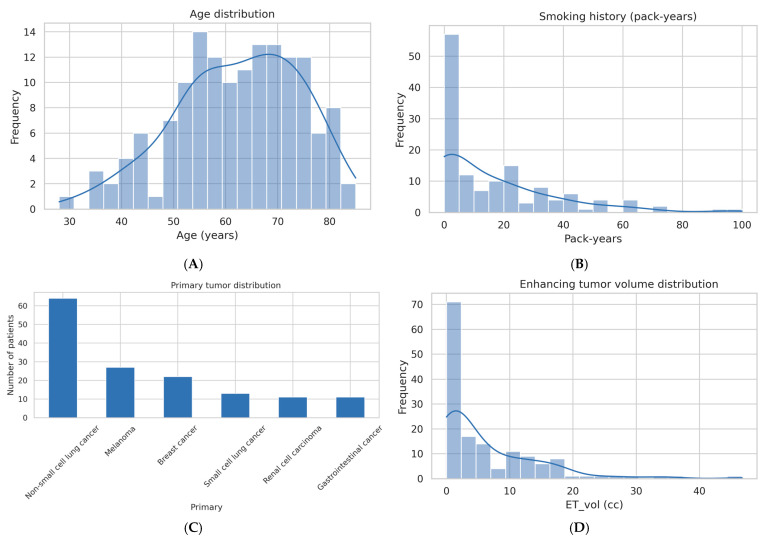
Patient demographics and baseline characteristics. (**A**) Distribution of patient age in the study cohort (*n* = 148). The cohort shows a broad age range (28–85 years), with a median age of 63 years, reflecting a typical brain metastases population. (**B**) Distribution of smoking history (pack–years). The data demonstrate a right-skewed distribution, with most patients having low-to-moderate smoking exposure, but notable outliers reaching up to 100 pack–years. (**C**) Primary tumor type distribution. Non-small cell lung cancer was the most frequent primary tumor (43.2%), followed by melanoma and breast cancer. (**D**) Distribution of enhancing tumor volume (ET_vol). Significant heterogeneity was observed, with a median volume of 2.81 cc and outliers exceeding 40 cc.

**Figure 2 diagnostics-15-01242-f002:**
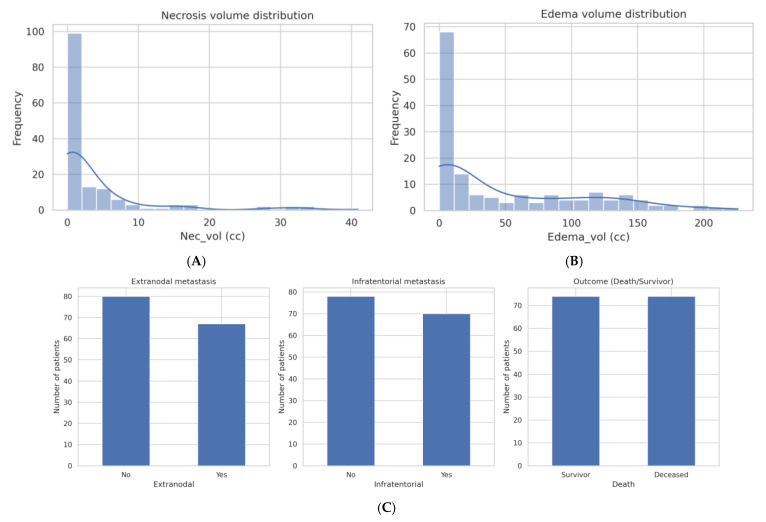
Tumor Characteristics and Clinical Outcomes. (**A**) Distribution of necrosis volume (Nec_vol). The majority of tumors exhibited minimal necrosis, although a subset demonstrated substantial necrotic volumes exceeding 30 cc. (**B**) Distribution of peritumoral edema volume (Edema_vol). Several patients showed extensive edema, with maximum volumes reaching up to 225.87 cc. (**C**) Presence of infratentorial metastases, extranodal metastases, and death outcomes, illustrated through proportional bar charts with balanced outcome distributions.

**Figure 3 diagnostics-15-01242-f003:**
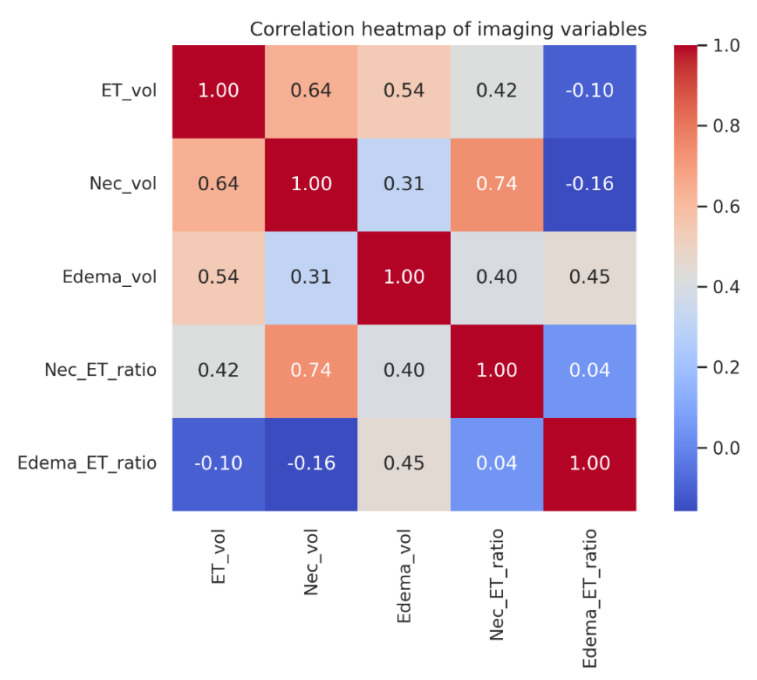
Correlation heatmap of continuous imaging variables (ET_vol, Nec_vol, Edema_vol, Nec_ET_ratio, Edema_ET_ratio). Moderate positive correlations were observed between edema volume and edema-to-tumor ratios (r = 0.45), indicating that larger edema volumes are generally associated with higher edema-to-tumor ratios, though this relationship is not linear or absolute.

**Figure 4 diagnostics-15-01242-f004:**
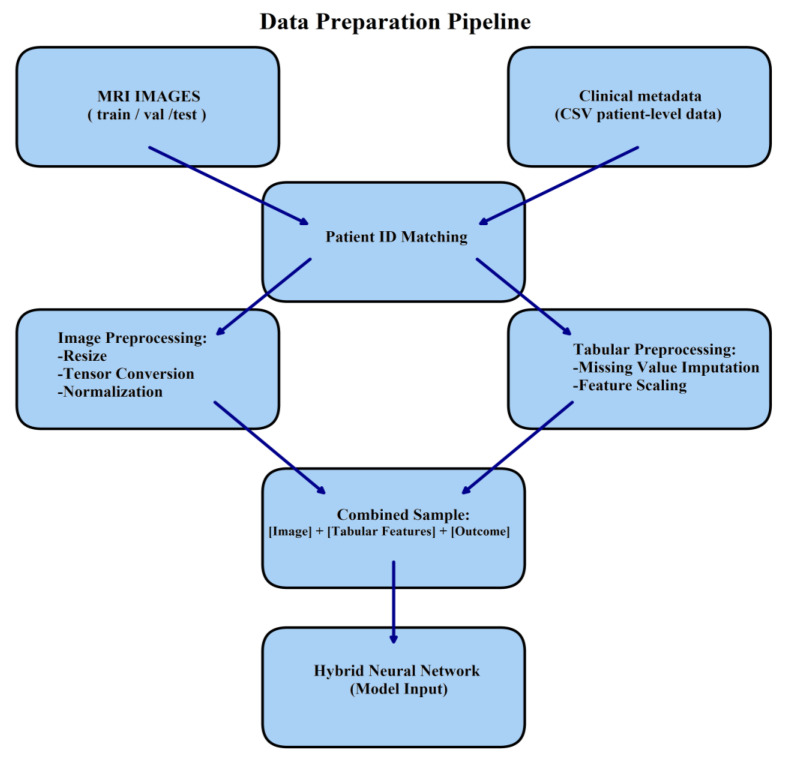
An overview of the dataset construction process. MRI slice images from training, validation, and testing folders are matched with patient-level clinical metadata via patient IDs. Each sample combines an MRI slice, associated tabular clinical features, and survival outcome labels. Images undergo preprocessing including resizing, tensor conversion, and normalization before being fed into the hybrid neural network models.

**Figure 5 diagnostics-15-01242-f005:**
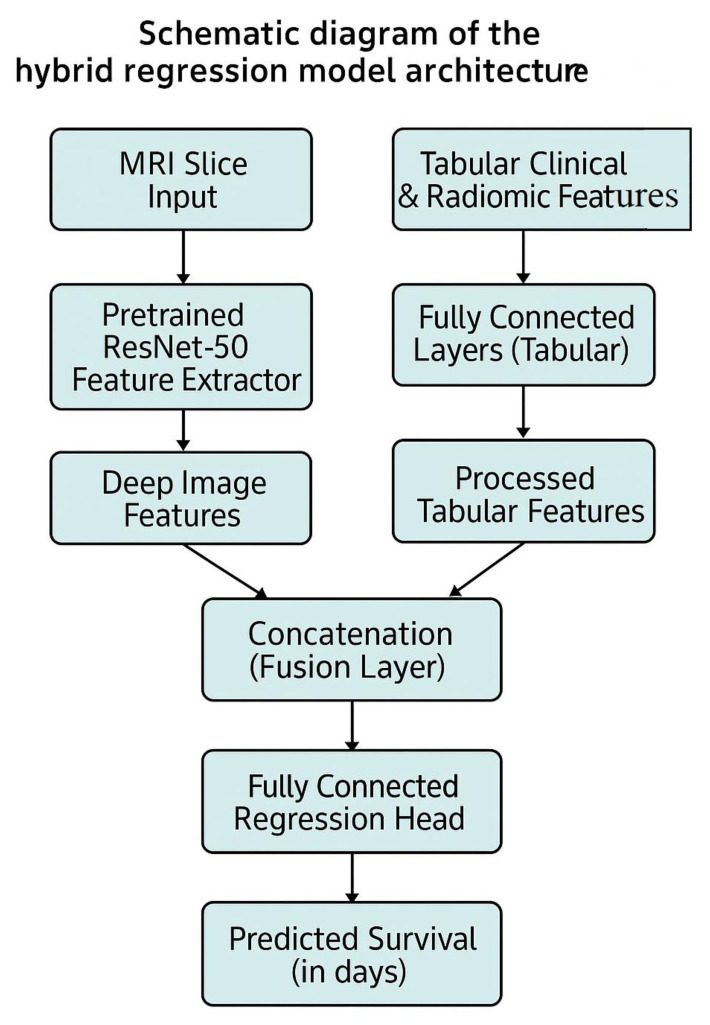
The model consists of two distinct input branches: (left) an imaging branch using a pretrained ResNet-50 backbone to extract deep features from MRI slices, and (right) a tabular branch composed of fully connected layers to process structured clinical and radiomic features. The outputs of these branches are concatenated and passed through a fully connected regression head, ultimately predicting patient survival in days. This multimodal fusion approach enables the model to leverage both imaging biomarkers and clinical variables for enhanced prognostic performance.

**Figure 6 diagnostics-15-01242-f006:**
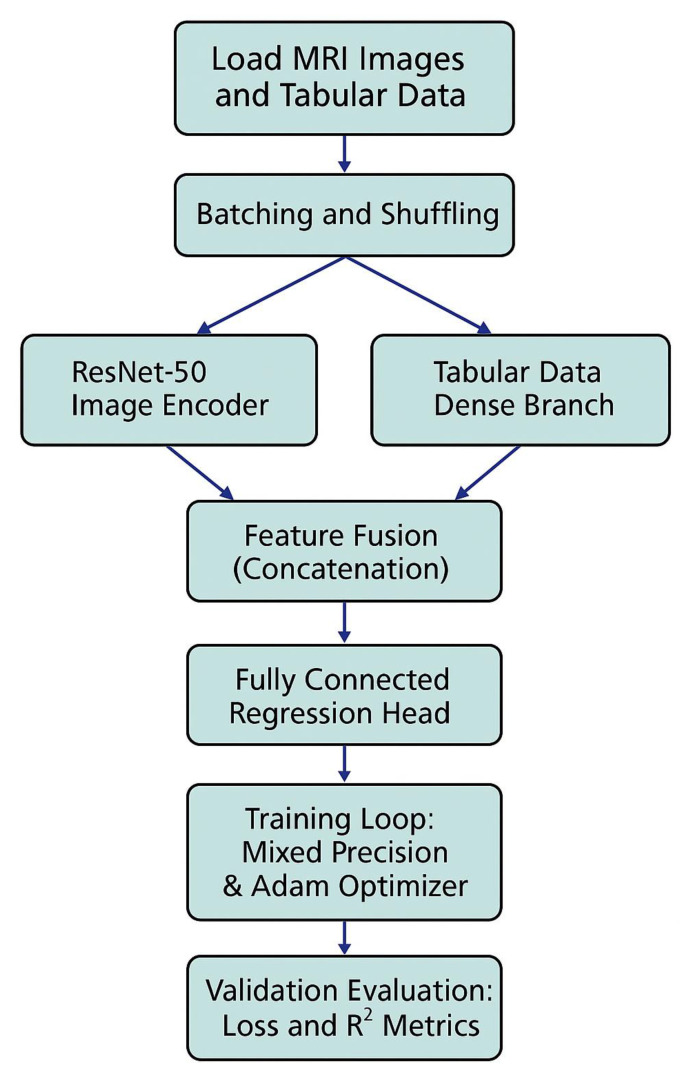
MRI images and associated tabular data are loaded, batched, and shuffled. Feature extraction is performed via a ResNet-50 image encoder and tabular data branch. Training uses mixed precision optimization with the Adam optimizer. Model evaluation is carried out on the validation set with loss and coefficient of determination (R^2^) metrics computation.

**Figure 7 diagnostics-15-01242-f007:**
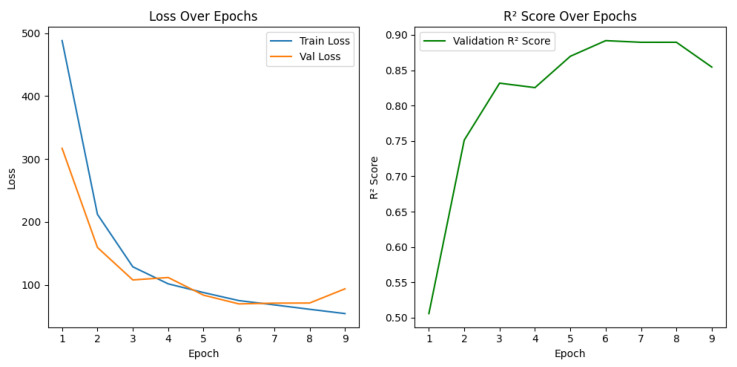
Training monitoring plots of the hybrid deep learning model. The **left panel** shows the training and validation loss curves over epochs, demonstrating progressive convergence and stability. The **right panel** illustrates the improvement in the validation R^2^ score throughout training, reaching optimal predictive performance before early stopping is triggered.

**Figure 8 diagnostics-15-01242-f008:**
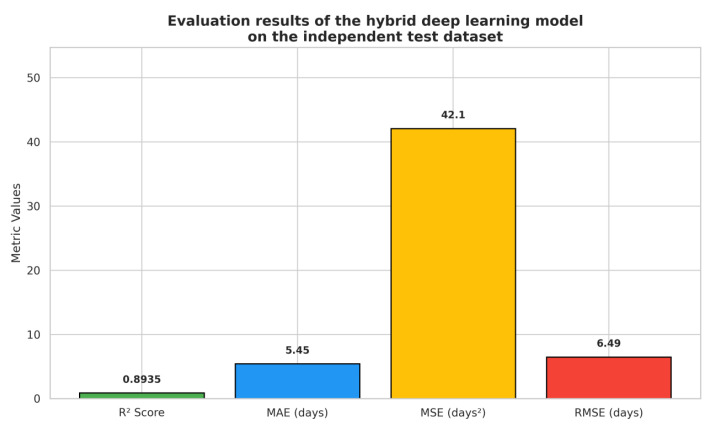
The figure summarizes regression performance metrics including the R^2^ score, mean absolute error (MAE), mean squared error (MSE), and root mean squared error (RMSE). The model achieved a high R^2^ score (0.8935) and a low MAE (5.45 days), highlighting strong predictive accuracy and clinical relevance. Although not shown in the plot, the Mean Absolute Percentage Error (MAPE) was also computed and found to be 43.58%. This value reflects the impact of short survival times in the dataset, where even small absolute errors yield high percentage deviations.

**Figure 9 diagnostics-15-01242-f009:**
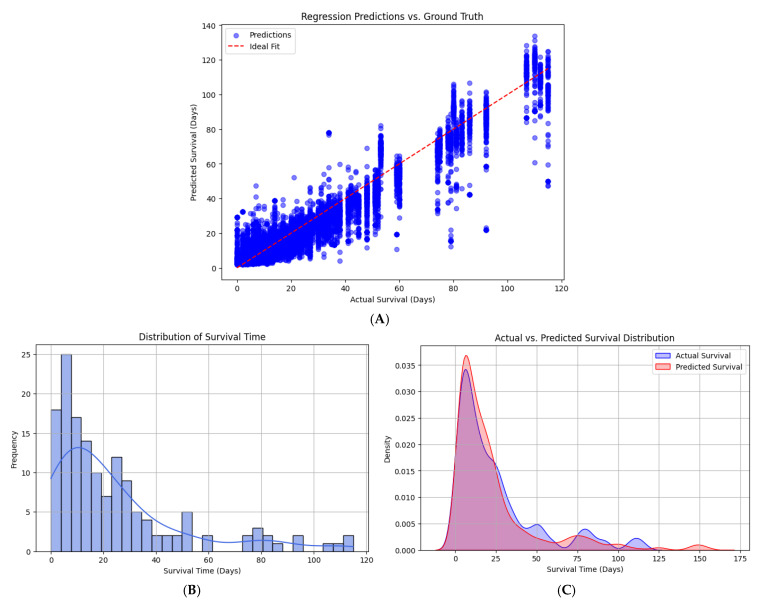
Model prediction correlation and survival time distributions. (**A**) Scatter plot showing the correlation between predicted and actual survival times for brain metastasis patients in the test set. The red dashed line represents the ideal line of perfect prediction. (**B**) Histogram with kernel density estimate (KDE) depicting the distribution of survival times in the study cohort. The data shows a right-skewed distribution, with most patients surviving less than 50 days. (**C**) Kernel density plots comparing actual versus predicted survival time distributions. The model successfully captures the main trend, although slight underestimation is observed for longer survival durations.

**Figure 10 diagnostics-15-01242-f010:**
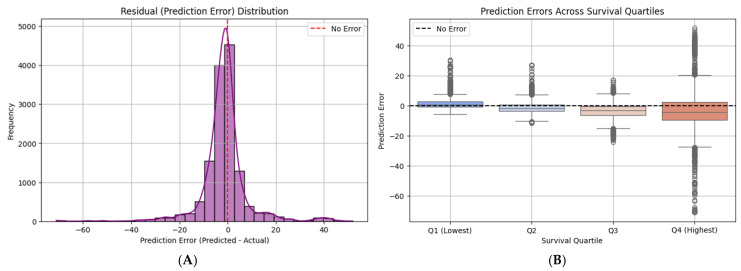
Analysis of prediction errors across the test cohort. (**A**) Histogram of prediction residuals (predicted minus actual survival times). Residuals are symmetrically distributed around zero, indicating no systematic bias, with most errors falling within a narrow and clinically acceptable range. (**B**) Boxplot of prediction errors stratified by survival quartiles. Prediction errors remain small and balanced in lower survival quartiles, while mild underestimation and increased variability are observed for patients with longer survival times (Q3 and Q4).

**Figure 11 diagnostics-15-01242-f011:**
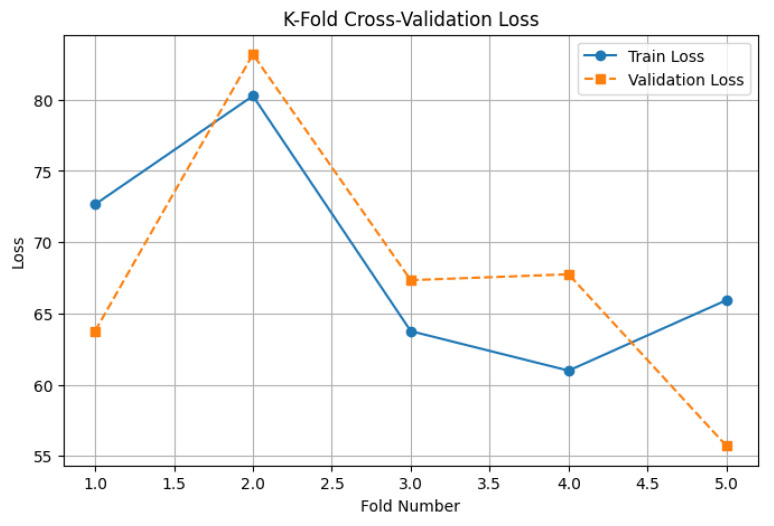
Training and validation loss across each fold during 5-fold cross-validation. The close tracking between train and validation losses demonstrates good generalization and model stability.

**Figure 12 diagnostics-15-01242-f012:**
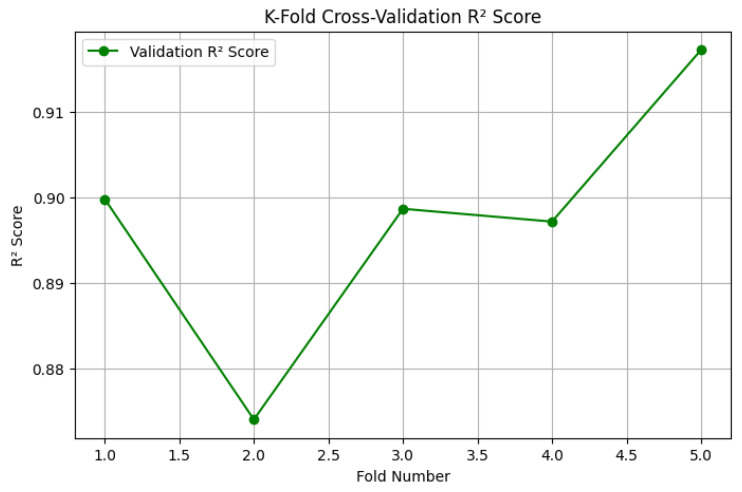
Validation R^2^ scores across the 5 folds in cross-validation. The stable and high R^2^ values indicate excellent predictive consistency and model generalization.

**Figure 13 diagnostics-15-01242-f013:**
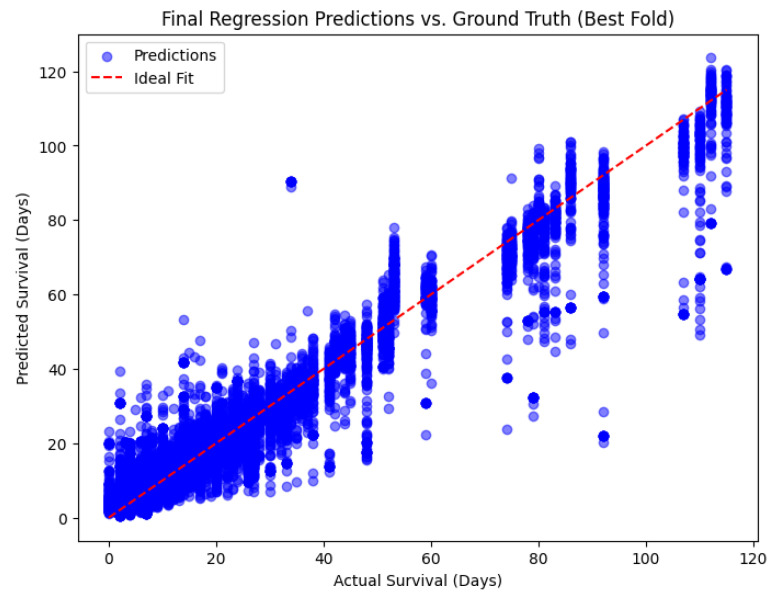
Final regression predictions versus ground truth for the best-performing fold model. Each point represents a single prediction from the model on the test dataset, compared to its actual survival outcome in days. The red dashed line indicates the ideal fit where predicted and actual values are equal. The dense clustering of points around this diagonal line highlights the model’s strong predictive performance and accuracy, supported by a final test R^2^ score of 0.9175.

**Figure 14 diagnostics-15-01242-f014:**
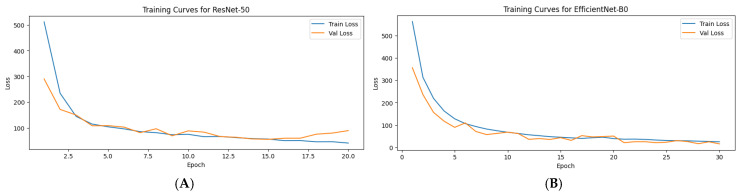
Training dynamics of hybrid deep learning models. (**A**) Training and validation loss curves for the Hybrid ResNet-50 model over 20 epochs. The model demonstrates steady convergence with a final validation R^2^ score of 0.91, although minor fluctuations after epoch 15 suggest possible mild overfitting or saturation of learning. (**B**) Training and validation loss curves for the Hybrid EfficientNet-B0 model over 30 epochs. The model exhibits smooth and consistent convergence, achieving a lower final validation loss and a high validation R^2^ score of 0.9759, indicating strong predictive performance and generalization capability.

**Figure 15 diagnostics-15-01242-f015:**
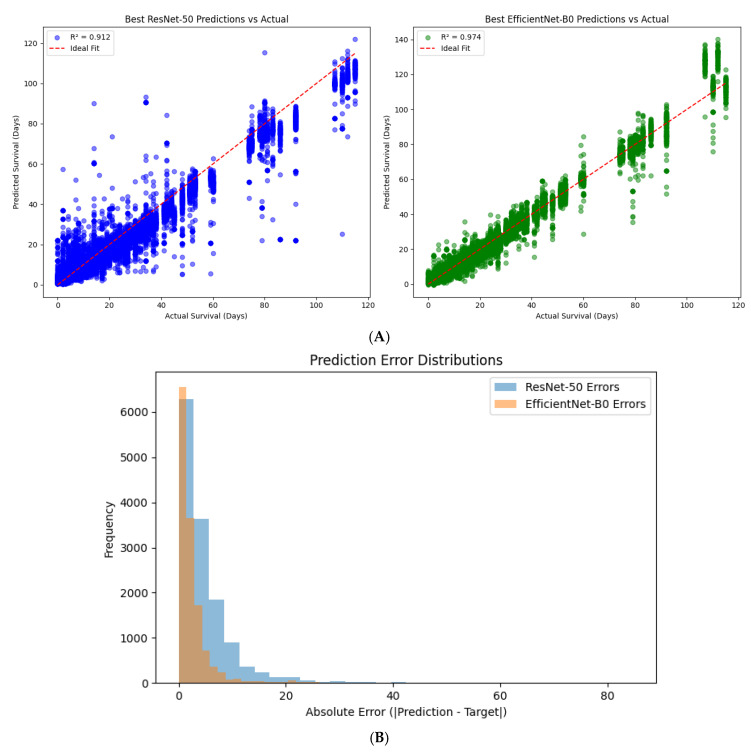
Comparative Model Performance of Hybrid ResNet-50 and EfficientNet-B0 Architectures. (**A**) Scatter plots comparing predicted versus actual survival times for the Hybrid ResNet-50 (**left**) and Hybrid EfficientNet-B0 (**right**) models on the test set. The red dashed line indicates the ideal prediction line; EfficientNet-B0 shows closer alignment with the identity line across the survival range. (**B**) Distribution of absolute prediction errors for both models. The EfficientNet-B0 model (orange) exhibits a narrower and more centered error distribution compared to ResNet-50 (blue), demonstrating superior prediction accuracy and reduced variability.

**Figure 16 diagnostics-15-01242-f016:**
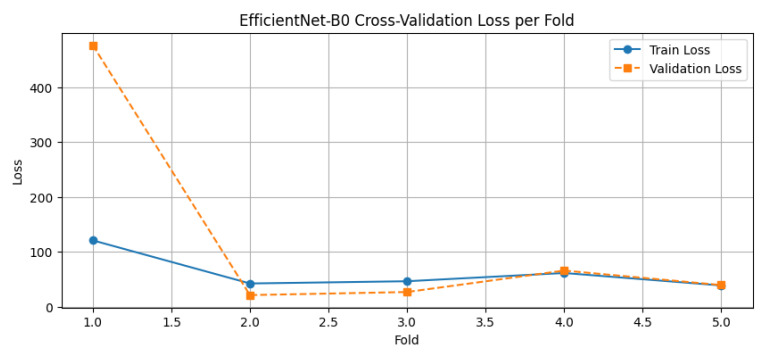
EfficientNet-B0 cross-validation loss per fold.

**Figure 17 diagnostics-15-01242-f017:**
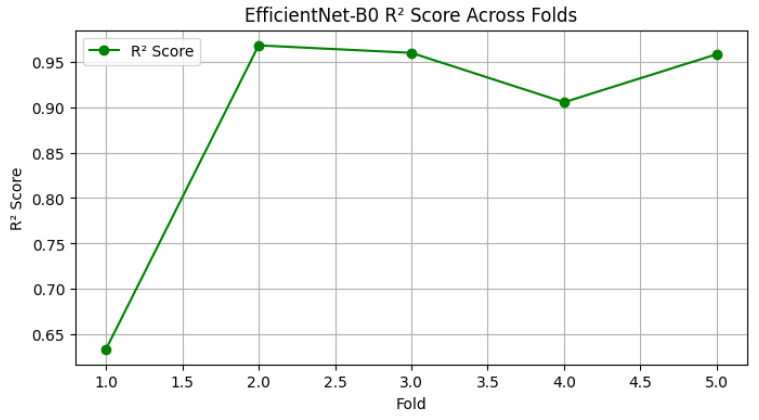
EfficientNet-B0 R^2^ scores across folds.

**Figure 18 diagnostics-15-01242-f018:**
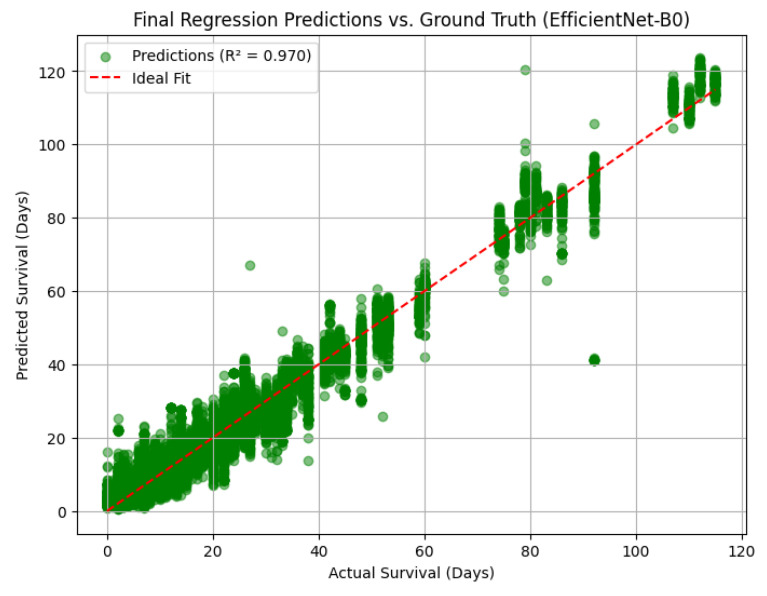
Predicted vs. actual survival for the best EfficientNet-B0 model.

**Figure 19 diagnostics-15-01242-f019:**
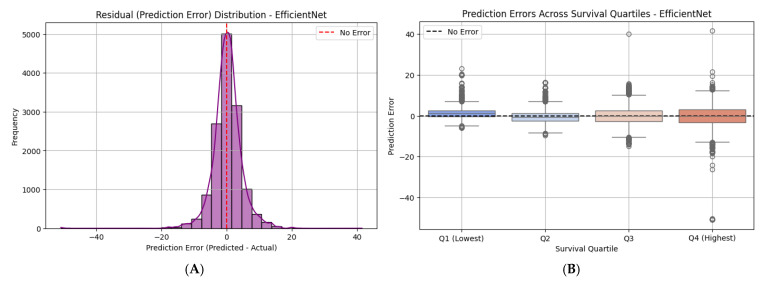
Residual and quartile error analysis for the hybrid EfficientNet-B0 model. (**A**) Distribution of prediction residuals (predicted minus actual survival time) centered tightly around zero, showing a near-normal distribution and minimal bias, with the majority of errors falling within ±10 days. (**B**) Boxplots of prediction errors across survival quartiles (Q1–Q4), demonstrating low median errors and consistent variance across all survival ranges, indicating robust model performance without systematic bias.

**Figure 20 diagnostics-15-01242-f020:**
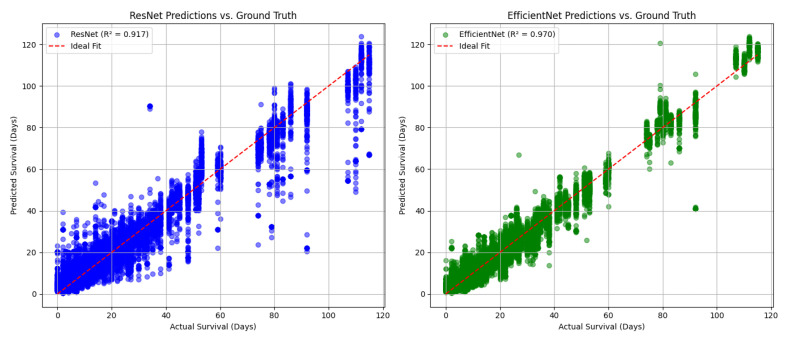
Comparative scatter plots of predicted vs. actual survival time for ResNet-50 (**left**) and EfficientNet-B0 (**right**). Predictions from the hybrid ResNet-50 model (**left**) show wider dispersion and more frequent underestimation, particularly at higher survival durations. The EfficientNet-B0 model (**right**) yields a tighter fit along the identity line, with higher overall accuracy (R^2^ = 0.970 vs. 0.917), confirming its superior predictive performance.

**Figure 21 diagnostics-15-01242-f021:**
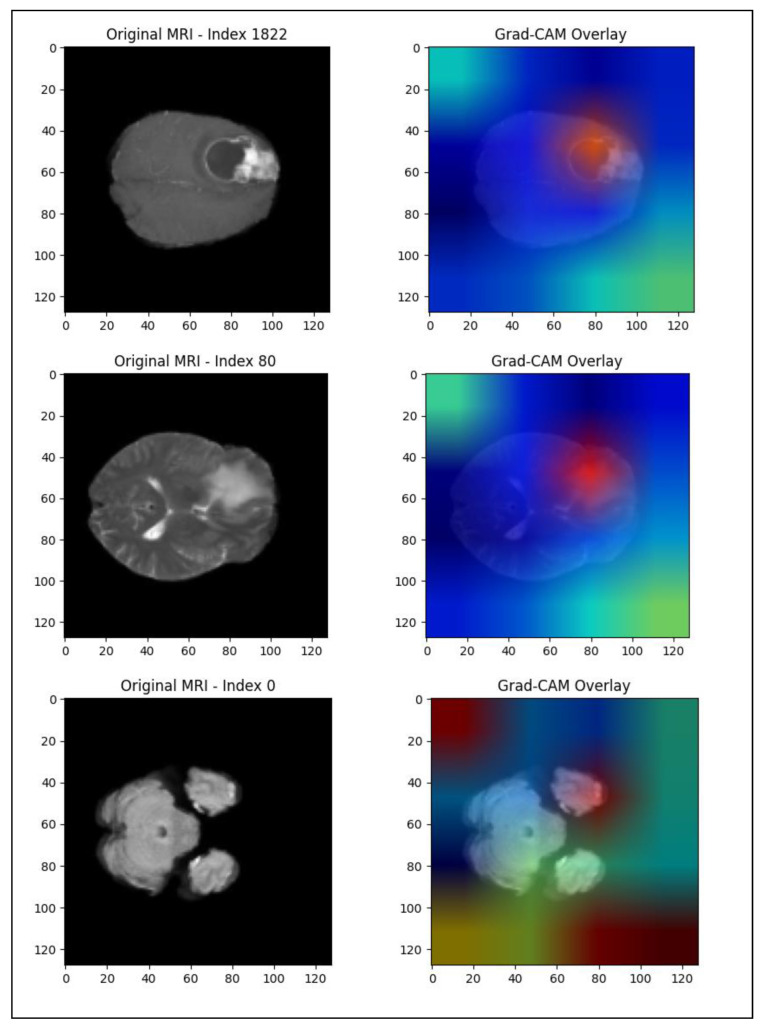
Example Grad-CAM visualization for model interpretability. The **left panel** shows the original MRI slice from the test dataset (Index 1822, 80, 0), highlighting a visible brain metastasis lesion. The **right panel** overlays the Grad-CAM heatmap, illustrating the regions most influential in the model’s survival prediction. The areas of highest model attention (in red) correspond to the tumor mass and surrounding peritumoral edema, aligning with known clinical prognostic indicators.

**Figure 22 diagnostics-15-01242-f022:**
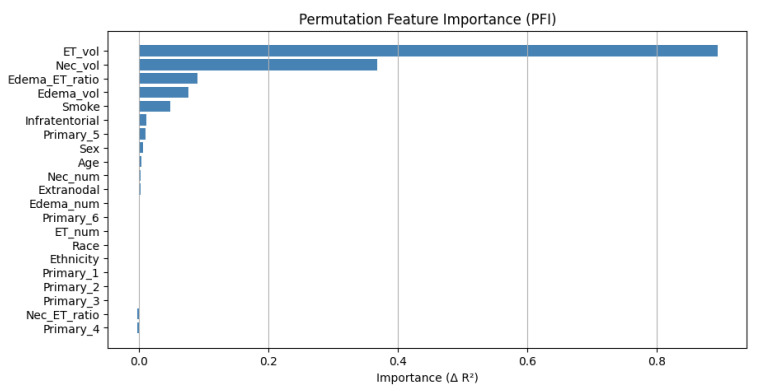
Permutation feature importance (PFI) analysis. Permutation feature importance (measured as a decrease in R^2^ score) for each input variable in the hybrid regression model. Enhancing tumor volume (ET_vol) was by far the most important feature, followed by necrosis volume and edema-related metrics. Features are ranked by descending ΔR^2^.

**Figure 23 diagnostics-15-01242-f023:**
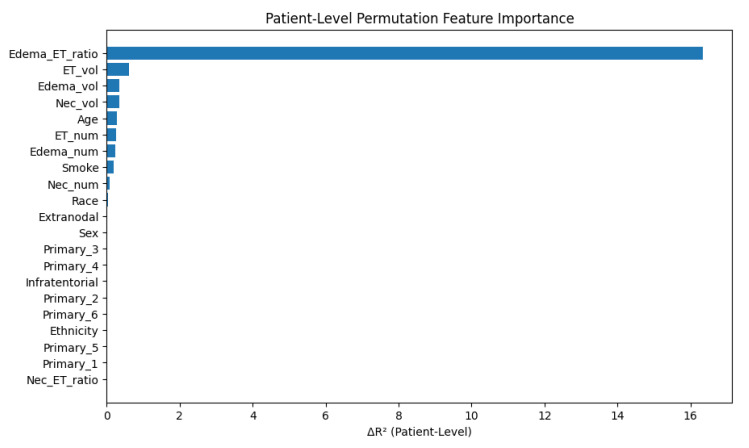
Patient-level permutation feature importance (PFI) analysis. Feature importance is quantified as the decrease in patient-level R^2^ score (ΔR^2^) upon random permutation of each feature. Edema_ET_ratio was the most influential predictor of survival, followed by volumetric metrics such as ET_vol and Edema_vol. Clinical and demographic features contributed modestly.

**Table 1 diagnostics-15-01242-t001:** Baseline patient and tumor characteristics.

Characteristic	Value
Total number of patients	148
Sex, n (%)	Female: 103 (69.6%); Male: 45 (30.4%)
Age, median (range)	63 years (28–85)
Smoking history (pack–years), median (range)	9 (0–100); missing in 8.8% of cases
Primary tumor site, n (%):	
— Breast cancer	22 (14.9%)
— Gastrointestinal cancer	11 (7.4%)
— Small cell lung cancer	13 (8.8%)
— Melanoma	27 (18.2%)
— Non-small cell lung cancer	64 (43.2%)
— Renal cell carcinoma	11 (7.4%)
Extranodal metastasis, n (%):	Yes: 67 (45.3%), No: 80 (54.1%), 1 missing
Infratentorial metastases, n (%):	Yes: 70 (47.3%), No: 78 (52.7%)
Survival time (days), median (range)	15.5 (0–115)
Enhancing tumor volume (ET_vol), median (range)	2.81 cc (0.01–46.73 cc)
Necrosis volume (Nec_vol), median (range)	0.57 cc (0–40.91 cc)
Edema volume (Edema_vol), median (range)	14.81 cc (0–225.87 cc)
Death outcome, n (%)	Deceased: 74 (50%), Survivors: 74 (50%)

**Table 2 diagnostics-15-01242-t002:** Cross-validation performance summary for each fold, including training loss, validation loss, R^2^ score, MAE, MSE, and RMSE.

Fold	Train Loss	Validation Loss	R^2^ Score	MAE	MSE	RMSE
1	72.66	63.78	0.9	4.78	63.84	7.99
2	80.27	83.18	0.874	5.54	83.27	9.13
3	63.76	67.34	0.899	4.78	67.41	8.21
4	60.99	67.75	0.897	4.77	67.82	8.24
5	65.94	55.72	0.917	4.45	55.78	7.47

**Table 3 diagnostics-15-01242-t003:** Comparative summary of selected studies on survival prediction in brain tumors using imaging and machine learning. Models differ in methodology, target formulation, and clinical population.

Study	Data Types Used	Methodology	Outcome Measure	Key Findings
Steyaert et al. [33]	MRI, clinical data, pathology	Multimodal deep learning fusion	Prognosis accuracy	Multimodal fusion improves outcome prediction in primary brain tumors.
Zhou et al. [34]	Multimodal MRI	Multi-channel CNN (M2Net)	Overall survival regression	Combining multiple MRI modalities improves survival estimation.
Banerjee et al. [35]	Radiomic features from MRI	Random forest model	Concordance index (C-index) = 0.82	Radiomics-based features are useful for prognosis in brain tumors.
Islam et al. [36]	MRI + clinical data	3D attention U-Net + radiomics	Survival estimation	Clinical and imaging features together yield more accurate predictions.
Chen et al. [37]	MRI radiomics	Multivariate Cox regression on radiomic features	Overall survival (regression and risk stratification)	Radiomic score significantly stratified patients by survival duration in NSCLC brain metastases
George et al. [38]	MRI radiomics	Random survival forest model	Progression-free survival and overall survival	Early imaging biomarkers predicted survival with high accuracy (C-index ~0.70–0.75)
Destito et al. [39]	MRI radiomics	Radiomics + classic ML classifiers	Overall survival and progression-free survival	Demonstrated feasibility of ML-based prediction in rare primary CNS lymphoma cases

Note: Due to methodological heterogeneity among studies—including variations in datasets, tumor types, survival targets (classification vs. regression), and evaluation metrics—formal statistical or meta-analytical benchmarking was not feasible. Instead, this table provides a structured, descriptive comparison focused on methodology, data modalities, and clinical relevance.

## Data Availability

Ramakrishnan et al. [24] through The Cancer Imaging Archive (TCIA).

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
