# Peer review of "Hybrid Deep Learning for Survival Prediction in Brain Metastases Using Multimodal MRI and Clinical Data"

_diagnostics, 2025, doi:10.3390/diagnostics15101242_

Round 1

Reviewer 1 Report (Previous Reviewer 1)

Comments and Suggestions for Authors

When the resubmitted article is examined, it is seen that the authors improved the article. I believe that the current revised form of the article is acceptable for publication.

Reviewer 2 Report (Previous Reviewer 2)

Comments and Suggestions for Authors

The revised manuscript demonstrates clear and substantial improvements in scientific quality, clarity, and structure. The authors have successfully addressed all prior reviewer comments, including those related to the novelty of the contribution, the integration of multimodal data, the interpretability of the model, and the limitations concerning dataset size and external validation. The methodology has been clearly refined, the results are robust and well-supported, and the discussion is now balanced and complete. The hybrid deep learning framework is well-justified and achieves high predictive accuracy, with clinical relevance and explainability. No further suggestions are needed at this stage. In its current form, the manuscript is scientifically sound, well-written, and suitable for publication. I commend the authors for their careful and thorough revisions.

Reviewer 3 Report (Previous Reviewer 3)

Comments and Suggestions for Authors

All requested changes have been made.

This manuscript is a resubmission of an earlier submission. The following is a list of the peer review reports and author responses from that submission.

Round 1

Reviewer 1 Report

Comments and Suggestions for Authors

In this manuscript, the authors propose a hybrid deep learning model for survival prediction in brain metastases by utilizing multimodal data, specifically MRI and clinical information. The study addresses a relevant and timely topic, especially considering the increasing interest in leveraging multimodal approaches for medical prognostication. It is evident that the authors have invested considerable effort in this work. Nevertheless, I have several observations and concerns that I believe should be addressed to strengthen the manuscript:

  1. Stated Contributions:
    The authors claim four key contributions: (i) a proposed multimodal fusion framework, (ii) use of a high-resolution dataset, (iii) enhanced performance and interpretability, and (iv) architectural benchmarking. However, the latter two—performance and benchmarking—are generally expected in comparative modeling studies and do not, in themselves, constitute novel contributions. Moreover, the use of a publicly available dataset, while commendable for reproducibility, does not represent a unique contribution. The core novelty appears to lie in the design of the hybrid multimodal deep learning model, but the manuscript should more clearly and appropriately articulate the originality and added value of the proposed approach.

  2. Data Distribution:
    The distribution of the clinical data, particularly age, appears non-uniform. For instance, Figure 1 suggests that the majority of samples fall within the 50–85 age range. A focused analysis within this range may yield less biased and more generalizable results. It would be beneficial for the authors to discuss the potential implications of this non-uniformity and consider evaluating model performance on a more balanced subset.

  3. Model Selection Justification:
    The selection of ResNet50 and EfficientNetB0 as the backbone architectures for the fusion model is not sufficiently justified. The manuscript would benefit from a discussion on the rationale behind choosing these particular models over others, including the specific characteristics (e.g., depth, parameter efficiency, transfer learning capabilities) that made them suitable for this application.

  4. Image Preprocessing:
    The decision to resize input images to 128×128 pixels requires further explanation. The authors should clarify whether this choice was driven by computational constraints, model architecture requirements, or dataset limitations. Furthermore, it is important to assess and discuss whether this downscaling led to a significant loss of relevant anatomical information that could affect model performance.

  5. Experimental Setup:
    Details about the computational environment used for training and evaluation (hardware specifications, GPU type, training duration, software libraries) are missing. Including this information is essential for reproducibility and for readers to evaluate the feasibility of deploying such models in practical settings.

  6. Model Overfitting:
    As illustrated in Figure 13, the training loss continues to decline after Epoch 7, while the validation loss begins to increase—indicative of potential overfitting. The authors should comment on this observation and describe any regularization techniques (e.g., dropout, early stopping) that were employed to mitigate overfitting.

  7. Architecture and Fusion Details:
    While a high-level block diagram of the proposed architecture is provided, the manuscript lacks sufficient detail about the integration of the SOTA models and the fusion mechanism itself. A deeper explanation of the fusion strategy (e.g., concatenation, attention-based, gating mechanisms) and how features from different modalities are combined would enhance the reader's understanding and the reproducibility of the work.

  8. Prediction Variability Across Age Groups:
    Figure 14 presents predicted survival days across different ages, showing a near-constant difference of approximately 20 days for each age group. This warrants further explanation. Additionally, it would be useful to know whether any regression fitting or smoothing techniques were applied to model the survival trend more realistically.

  9. Figure Layout and Formatting:
    The figures throughout the manuscript occupy significant space and could be better organized to enhance readability. Reformatting them into multi-panel layouts (i.e., using rows and columns) would improve visual clarity and allow the authors to present more information without overwhelming the reader.

Reviewer 2 Report

Comments and Suggestions for Authors

Thank you for your submission. While your work shows strong technical execution and uses a real multimodal dataset, the manuscript lacks sufficient originality to meet the standards of a scientific journal publication in its current form. Below are specific observations and suggestions:

1. On Contributions and Novelty – Pages 3, 38
You claim a novel contribution by using regression instead of classification for survival prediction. However, this approach has already been explored in prior works, such as Islam et al. and Zhou et al., which are also cited by you in Table 3 (page 38). Therefore, this framing is not adequate to support novelty​.

Additionally, the architecture used (EfficientNet-B0) is not customized or extended. It is used as-is without modification. The fusion between image and clinical data is done via simple concatenation, as described in Section 3 (page 13), which is a commonly adopted method in medical AI applications​.

2. On Dataset – Pages 5–7
While your dataset includes a high number of MRI slices, the patient cohort is limited to 148 cases (Table 1, page 6). This is relatively small for training deep learning models and raises concerns about generalizability. Furthermore, no external validation is included, and your generalizability claim is stated hypothetically on page 37:

“further validation on external, multi-institutional datasets will be necessary…”​

3. On Results and Figures – Pages 24–34
Table 2 (Page 24): The cross-validation metrics are strong, with R2 ranging from 0.874 to 0.917. However, they are not contextualized against prior benchmarks in the same domain.

Figure 28 (Page 29): The predicted vs actual survival plot shows tight clustering, but this is expected in regression tasks with small datasets. It does not alone prove robustness.

Figures 31 & 32 (Pages 32–33): These show visual comparisons and Grad-CAM examples. While informative, they rely on standard tools. The visual analysis does not present an innovation in interpretability.

4. On Interpretability – Pages 33–34
Your use of Grad-CAM and Permutation Feature Importance (PFI) is correctly implemented, but both tools are established and widely adopted in AI. Their application in this work is descriptive, not methodological. Section 5.2 (Page 34) states:

"Enhancing tumor volume emerged as the most critical predictor..."
This finding is expected and aligns with known literature, not presenting a surprising insight​.

5. On Comparison with Previous Work – Page 38
In Table 3, your comparisons to other studies are mostly descriptive. There is no statistical or meta-analytical comparison that distinguishes your method’s performance meaningfully from those cited​.

Suggestions to Improve the Work
To elevate your study and make it suitable for a future submission:

- Introduce a genuine scientific contribution --- whether a novel fusion mechanism, custom CNN design, uncertainty quantification, or new explainability framework.

- Benchmark your results statistically against prior works using shared datasets or strong baselines.

- Justify your dataset size and add external validation to improve the model's generalizability claims.

- Refine your interpretation of results — emphasize insights beyond what the architecture already reveals (e.g., deeper analysis of model failure cases).

** In summary, the manuscript presents a clean technical application but lacks the scientific depth and innovation required for publication. I encourage you to reconsider the core contribution and methodology before resubmitting.

Reviewer 3 Report

Comments and Suggestions for Authors

The study proposes a novel hybrid deep learning framework to predict overall survival by integrating volumetric MRI-derived imaging biomarkers with structured clinical and demographic data. The study contributes to the literature in terms of subject matter and results obtained. However, some minor changes should be made.

1. Start by explaining what will be explained here instead of moving from main headings to direct subheadings.

2. Write the paper organization in one paragraph.

3. Improve the readability of the text in Figure 11.

4. The metrics in the model evaluation section are known to researchers. It is recommended to give them as a table in the appendix instead of describing them.

5. Although the studies on the subject have been extensively researched, only 4 studies are included in the table in the discussion section and a comparison is made. Please include more studies in the table.

6. Fit Table 2 and Table 3 on the page. Also, in Table 3, the “data types used” and “outcome measure” of the study are not included.

7. Check the numbering (double numbering) in the references.

8. References are not written in the same format. Please revise and correct them (e.g. reference 41 date in bold, reference 24, blue color).

Reviewer 4 Report

Comments and Suggestions for Authors

- The approach proposed for survival prediction for patients with brain metastases using MRI and clinical data is valid and the results might be promising; however there is a major concern involving data usage, and the manuscript is rather limited both in terms of organisation and contents, not currently meeting publication standards;

- It is claimed that one of the contribution is a "High Resolution Dataset", what is not obvious from the  paper, especially when publicly available data was used;

- Also, "Architecture benchmarking" is to be seen as a part of the process to reach the best performing pipeline, and not a key finding itself;

-  Background and the state of the art should serve to frame the field and research questions, and therefore should appear in, or right after, "Introduction" not in the scope of performance analysis (section 6 in the case);

- Balancing the dataset by removing part of its data is severely biasing process which, at the end, means that the methodology is not tuned / assessed in real world data (which is naturally unbalanced) and this is a critical limitation;

- Authors report a "strong correlation" with r=0.64 (line 291), and although acknowledging a significant figure,  "strong" is usually used only with r > 0. 70;

- The BraTS-MET in mentioned in line 296 with not previous information about it (or its use), and only referenced later in line 1104 (and actually in an odd formatting style);

- The term "PyTorch dataset" (326) is vague should be clarified;

- Figure 9 and its caption should be on the same page;

- In Figures 9 and 10, the text in the boxes should be made more readable (higher size font);

- No detailed  information is provided about the FCs modules that process clinical / radiomics data; it would therefore be impossible for other researchers to reproduce;

-  "Computational efficiency" is mentioned (line 388), but no indicators are presented (complexity, FLOPS, GMACS, etc);

- The data set split (training / validation / test) is not provided;

- Training details (curves, etc) should be presented ahed of the results;

- Still about training, which layers were trained and which were frozen?

- Concerning the results in Figure 9, the MAE of 5.45 days  needs be discussed in relative terms (associated MAPE);

- In terms of explainability, no information is provided on how backpropagation is considered on the part of the pipeline that deals with clinical / radiomics data. 
